# Dialogue Chain-of-Thought Distillation for Commonsense-aware Conversational Agents

**Hyungjoo Chae**[1*]  **Yongho Song**[1*]  **Kai Tzu-iunn Ong**[1]
**Taeyoon Kwon**[1]  **Minjin Kim**[1]  **Youngjae Yu**[1]
**Dongha Lee**[1]  **Dongyeop Kang**[2]  **Jinyoung Yeo**[1]

Yonsei University[1]    University of Minnesota[2]
{mapoout, kopf_yhs, yjy, donalee, jinyeo}@yonsei.ac.kr    dongyeop@umn.edu

## Abstract

Human-like chatbots necessitate the use of commonsense reasoning in order to effectively comprehend and respond to implicit information present within conversations. Achieving such coherence and informativeness in responses, however, is a non-trivial task. Even for large language models (LLMs), the task of identifying and aggregating key evidence within a single hop presents a substantial challenge. This complexity arises because such evidence is scattered across multiple turns in a conversation, thus necessitating integration over multiple hops. Hence, our focus is to facilitate such multi-hop reasoning over a dialogue context, namely dialogue chain-of-thought (CoT) reasoning. To this end, we propose a knowledge distillation framework that leverages LLMs as unreliable teachers and selectively distills consistent and helpful rationales via alignment filters. We further present DOCTOR, a **D**ial**O**gue **C**hain-of-**Th**ought **R**easoner that provides reliable CoT rationales for response generation[1]. We conduct extensive experiments to show that enhancing dialogue agents with high-quality rationales from DOCTOR significantly improves the quality of their responses[2].

## 1   Introduction

Commonsense reasoning is crucial in human conversation (Richardson and Heck, 2023). However, most conversational agents still lack commonsense reasoning, limiting their capability to engage in rich conversations with users (Arabshahi et al., 2021). Recent studies (Gao et al., 2022; Zhou et al., 2022a,b) aim to tackle this issue by generating commonsense knowledge (Hwang et al., 2021; Bosselut et al., 2019) relevant to the dialogue context, but

---

*Equal contribution

[1]We release our source code on https://github.com/kyle8581/DialogueCoT.

[2]We release demonstrations of dialogue CoT reasoning in https://dialoguecot.web.app/.

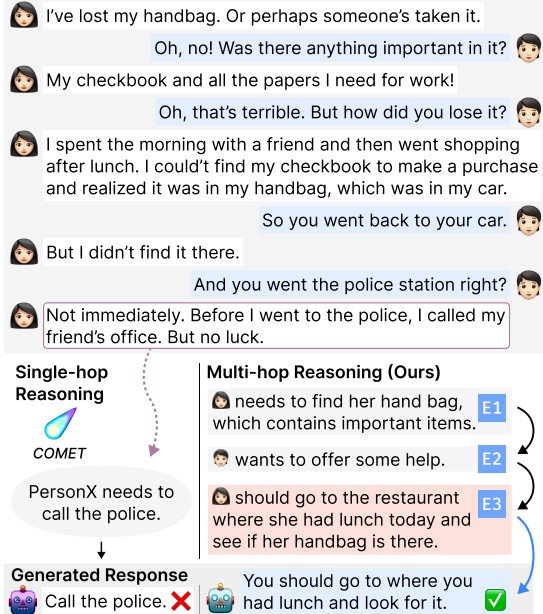

Figure 1: Comparison between responses generated via single-hop reasoning and multi-hop reasoning.

they still suffer from limited or incorrect reasoning that leads to dull and incoherent responses, as shown on the left in Figure 1.

The problem lies in that commonsense reasoning in a conversation involves multi-hop reasoning. Since key evidence is scattered across and implicitly stated in multiple dialogue turns (Liu et al., 2021a; Zhao et al., 2022), it is challenging to capture all necessary information in a single hop. For example, the response on the right in Figure 1 (*e.g.*, "*You should go to ...*") can only be obtained by integrating multiple implicit evidence (*e.g.*, E1, E2, and E3) from the dialogue. The process of finding and aggregating such scattered evidence takes multiple steps, highlighting the need for multi-hop reasoning for coherent and informative responses.

Inspired by the success of Large Language Models (LLMs) (Brown et al., 2020), we formulate multi-hop commonsense reasoning as Chain-of-Thought (CoT) reasoning (Wei et al., 2022) for dia-

logue response generation, namely **Dialogue CoT**. Our goal is to facilitate dialogue CoT reasoning by enabling language models to decompose commonsense reasoning into multiple steps and generate *rationale* as a sequence of inferred commonsense knowledge required for response generation.

Despite their potential effectiveness, we observe two limitations of prompting LLMs for commonsense reasoning in conversations: (1) LLMs tend to rely much on explicit cues, necessitating task-specific constraints to seek implicit knowledge. (2) LLMs exhibit poor *alignment* between rationales and dialogues, resulting in inconsistent and unhelpful rationales. These challenges motivate the need for a robust symbolic distillation mechanism (West et al., 2022; Kim et al., 2022a) that selectively transfers CoT capabilities from LLMs to train a reliable CoT reasoner.

Our contributions are threefold: (1) We propose a dialogue chain-of-thought distillation framework that extracts plausible rationales from unreliable LLMs and collects high-quality CoT rationales via *iterative question answering* and *alignment filtering*. (2) Using our framework, we collect 🍩 DONUT, a dialogue dataset annotated with high-quality CoT rationales. Our qualitative analysis of the collected rationales shows the effectiveness of our method for controlling the reliability of LLMs in extracting rationales. (3) With DONUT, we train DOCTOR, a **D**ial**O**gue **C**hain-of-**T**hought c**O**mmonsense **R**easoner that integrates implicit information in dialogue into rationale for generating responses[3]. We conduct experiments on response generation tasks to show that augmenting dialogue models with high-quality rationales from DOCTOR significantly improves their performance.

## 2 Dialogue Chain-of-Thought Reasoning

### 2.1 Preliminaries

Recent work (Wu et al., 2020; Zhou et al., 2022a,b; Gao et al., 2022; Zhong et al., 2021) aims to enrich dialogue modeling by augmenting dialogue agents with commonsense knowledge to infer implicit information in conversations. Specifically, a dialogue model $\theta$ is given commonsense knowledge $Z$ as an additional input to predict the next response $u_t$ for the dialogue context $U_{<t}$ of $t-1$ turns:

$$u_t \sim P_\theta(\cdot | Z, U_{<t}) \tag{1}$$

In these approaches, commonsense knowledge $Z$ is either retrieved from symbolic knowledge bases (KBs) (Zhou et al., 2022b; Gao et al., 2022) such as ATOMIC (Hwang et al., 2021), or generated from neural KBs such as COMET (Bosselut et al., 2019). These methods, however, tend to miss subtle yet implicit details in a conversation (Shwartz et al., 2020; Schlegel et al., 2022), leading to dull and incoherent responses. We posit that commonsense reasoning in a conversation requires multiple hops to capture such implicit details scattered across multiple turns (Liu et al., 2021b; Zhao et al., 2022).

### 2.2 Formulating Chain-of-Thought Reasoning in Dialogues

Inspired by the success of rationale-augmented LLMs on multiple reasoning tasks (Wei et al., 2022; Zhou et al., 2023), we formulate multi-hop reasoning in conversation as *dialogue CoT reasoning*, which decomposes reasoning into multiple steps and combines inferred commonsense knowledge into a *rationale* that supports response generation. With dialogue CoT reasoning, dialogue agents can generate coherent responses by identifying relevant contextual cues in a dialogue and making use of implicit information underlying the context.

A naive approach to facilitate dialogue CoT reasoning is to apply CoT prompting on LLMs. However, we find this approach is suboptimal due to the following limitations: (1) LLMs attend more to explicit cues (*e.g.* lexical overlap) in dialogues for reasoning, requiring task-specific constraints to guide the model to infer implicit information; (2) The rationales are often misaligned with the dialogues, *i.e.*, inconsistent with the contexts (Peng et al., 2023) or unhelpful in response generation (Jung et al., 2022). Based on these insights, we aim to construct a reliable CoT reasoner that generates high-quality rationales for dialogue agents.

## 3 Dialogue Chain-of-Thought Distillation

In this section, we propose a robust knowledge distillation framework that extracts plausible CoT rationales from an unreliable LLM (§3.1) and selectively distills high-quality rationales via alignment filters (§3.2) to (re-)train a reliable CoT reasoner. Figure 2 presents an overview of our framework.

### 3.1 QA-driven Rationalization

Our framework is designed to augment existing large-scale dialogue corpora with dialogue CoT

---

[3]We release the model checkpoint on https://huggingface.co/DLI-Lab/DOCTOR.

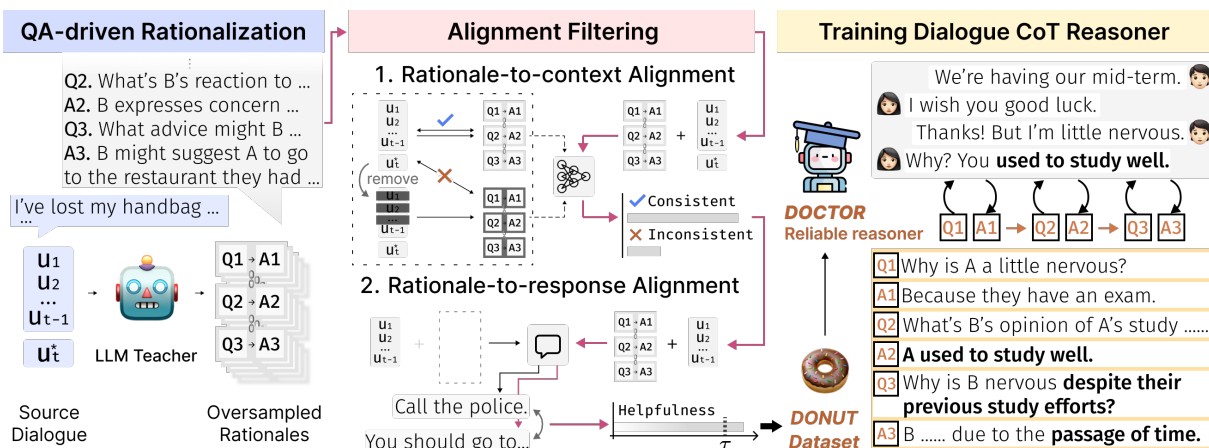

Figure 2: **Overview of our framework.** We leverage an LLM to collect CoT rationales and apply filters to selectively annotate them. The same dialogue from Figure 1 is used to showcase rationale generation (left) and alignment filtering (middle). The dotted square shows the training of the critic model with counterfactual rationales.

rationales by leveraging the capability of LLMs to rationalize. We first prompt an LLM to generate a plausible rationale $Z^*$ for a dialogue context $U_{<t}$ and the ground-truth response $u_t$ such that the next response $u_t$ is induced from the rationale $Z^*$:

$$Z^* = \underset{Z}{\operatorname{argmax}} \, P_{\text{LLM}}(Z|u_t, U_{<t}) \qquad (2)$$

Specifically, we represent a rationale $Z$ with a sequence of $k$ question-answer pairs $\{(q_i, a_i)\}_{i=1}^k$, where $q_i$ is an information-seeking question about implicit information $a_i$ in $U_{<t}$. By instructing an LLM to iteratively generate questions and answers, we ask the model to pinpoint relevant contextual cues and infer underlying knowledge that supports response generation.

In practice, we choose a set of commonsense relations from ATOMIC (Hwang et al., 2021) that are commonly used in dialogue domains. We prompt LLMs to construct questions $q_i$ based on the relation type to guide the model to construct questions pertaining to conversations. We further include 5 demonstrations of dialogue CoT, each of which contains human-authored question-answer pairs for a dialogue $U = [U_{<t}; u_t]$. We present the list of commonsense relations along with the example prompt used for rationale collection in Appendix A.1.

## 3.2 Alignment Filtering

To ensure the quality of the annotated rationales, we further introduce two *alignment filters* that filter out rationales based on their alignment with the dialogue contexts and the ground-truth responses.

**Rationale-to-context alignment.** LLMs tend to hallucinate facts without attending to the con-

text (Peng et al., 2023), which can often lead to rationales that are misaligned with the dialogue context. Inspired by West et al. (2022), we minimize such inconsistent rationales from LLMs by employing a critic model to detect *counterfactual* rationales generated without correctly grounding on the dialogue context[4]. We ask the LLM to generate a counterfactual rationale $\tilde{Z}$ from a counterfactual context $\tilde{U}_{<t}$ containing only the last utterance:

$$\tilde{Z} = \underset{Z}{\operatorname{argmax}} \, P_{\text{LLM}}(Z|u_t, \tilde{U}_{<t}) \qquad (3)$$

The critic model is trained to distinguish between $Z^*$ and $\tilde{Z}$ for given dialogue contexts. We sample 6K dialogues from SODA (Kim et al., 2022a) and collect 6K $(U_{<t}, Z^*)$ pairs by manually choosing consistent $Z^*$ from the set of generated rationales. We then construct 6K $(\tilde{U}_{<t}, \tilde{Z})$ pairs for the collected samples, resulting in 5k training instances of $(U_{<t}, Z^*)$ and $(\tilde{U}_{<t}, \tilde{Z})$ pairs for our critic model.

**Rationale-to-response alignment.** We consider a rationale to be aligned with a response if augmenting dialogue models with the rationale helps predicting the ground-truth response. Hence, we introduce an indicator function helpful($\cdot$) to determine if a dialogue model $\theta$ benefits from a rationale $Z$ when predicting the ground-truth response $u_t$ given a context $U_{<t}$[5]. Formally, we define a boolean function helpful($\cdot$) as:

$$\texttt{helpful}(Z) = \mathbb{1}\left[\frac{P_\theta(u_t|Z, U_{<t})}{P_\theta(u_t|U_{<t})} > \tau\right] \qquad (4)$$

---

[4]We implement the critic model with RoBERTa-large (Liu et al., 2019). See Appendix A.3 for more details.

[5]We use Cosmo-3B (Kim et al., 2022a) trained on a large-scale dialogue dataset covering diverse social interactions.

where $\mathbb{1}[\cdot]$ is a binary indicator and $\tau$ is a hyperparameter[6]. Intuitively, higher probability $P_\theta(u_t|Z, U_{<t})$ indicates that the rationale $Z$ is more helpful in predicting the response $u_t$.

## 3.3 Training DOCTOR

Using the annotated dialogue corpus, we train a **D**ial**O**gue **C**hain-of-**Th**ought **R**easoner, namely DOCTOR. We train our model with a causal language modeling objective to predict the probability of generating the rationale $Z^*$ given the dialogue history $U_{<t}$. Essentially, the training objective can be formulated as next-token prediction over a sequence of question-answer pairs $(q_i, a_i)$ in a rationale, where the model iteratively predicts $q_i$ and $a_i$ following previously generated question-answer pairs. We posit that by learning to generate and answer subquestions, the model can identify all implicit information required to infer the corresponding commonsense knowledge from the dialogue.

DOCTOR is built on top of OPT-1.3B (Zhang et al., 2022) and is trained using 80% of the annotated data with a constant learning rate of 5e-4 for 5 epochs. See Appendix A for details.

## 4 🍩 DONUT

Alongside DOCTOR, we present its training corpus, 🍩 DONUT, a **D**ial**O**gue chai**N**-of-tho**U**ght datase**T** with annotated rationales for dialogue CoT reasoning[7]. We choose three human-collected dialogue datasets, DailyDialog (Li et al., 2017), DREAM (Sun et al., 2019), and MuTual (Cui et al., 2020), to sample source dialogues for annotation[8]. We also include 5% of the dialogues in SODA (Kim et al., 2022a), a million-scale social dialogue dataset, for scalability. In total, we obtain 10K dialogues for annotation. For each utterance in a dialogue (except for the first one), we instruct ChatGPT to generate 10 rationale candidates.

Using the two alignment filters from §3.2, we filter out 122,319 candidates (24.98%) that are either inconsistent with the dialogue context or not helpful in predicting the next response. The resulting dataset, 🍩 DONUT, consists of 10K dialogues with 367K CoT rationales. Table 1 shows a sample from

---

[6]We use 0.95 for $\tau$. We provide the distribution of the ratio in Appendix D.1.
[7]DONUT is available on `https://huggingface.co/datasets/DLI-Lab/DONUT`.
[8]We use a subset of dialogue samples from the three datasets as curated by Ghosal et al. (2022a).

---

**Dialogue Context**:

**A:** Hi, Viggo. How are you doing today?

**B:** Hey, Yovani. I'm doing all right. Thanks for asking.

**A:** No problem. I saw that you left your coffee mug on the counter this morning. Did you forget to take it with you?

**B:** Yeah, I did. Thanks for grabbing it for me.

**A:** No problem at all. I know how busy you are and I didn't want you to have to come back for it later.

**B:** You're a lifesaver, Yovani. Seriously, thank you so much.

**Dialogue Chain-of-Thought Rationale**:

**Q1:** What did Person A do for Person B? (oReact)

**A1:** Person A grabbed Person B's coffee mug for him when he forgot it.

**Q2:** What is Person B's reaction to Person A's help? (xReact)

**A2:** Person B is thankful and expresses gratitude to Person A for helping him out.

**Q3:** What might Person A want to convey to Person B, based on their previous interactions? (xIntent)

**A3:** Based on their previous interactions, Person A might want to convey that he understands how important coffee is to Person B and that he is always willing to help him out.

**Ground-truth Response**:

• **A**: Any time. I know how much you love your coffee in the morning.

Table 1: A sample from 🍩 DONUT.

| Description | DONUT | CICERO | Reflect-9K | ComFact |
|---|---|---|---|---|
| # Dialogues | 10K | 5.6K | 600 | 769 |
| # Turns with Inf. | 46K | 28K | 600 | 6K |
| # Inferences | 367K | 53K | 9K | 52K |
| Avg. # words per Inf. | 78.6 | 12.0 | 5.4 | 3.4 |

Table 2: Statistics of DONUT vs. human-authored dialogue datasets with annotated commonsense knowledge.

DONUT. Further analyses on the alignment filtering and generated rationales are in Appendix D.

## 4.1 DONUT vs. Human-annotated Datasets

Here we summarize the advantages of DONUT over three dialogue datasets: CICERO (Ghosal et al., 2022a), Reflect-9K (Zhou et al., 2022a), and ComFact (Gao et al., 2022). These datasets provide high-quality human-annotated commonsense knowledge for dialogues.

**Large scale.** As shown in Table 2, DONUT contains a larger amount of annotated dialogue samples compared to existing dialogue datasets with human-annotated commonsense knowledge.

**Cost & time-efficiency.** Unlike human-authored datasets, DONUT is automatically annotated via ChatGPT in a time and cost-efficient manner. With

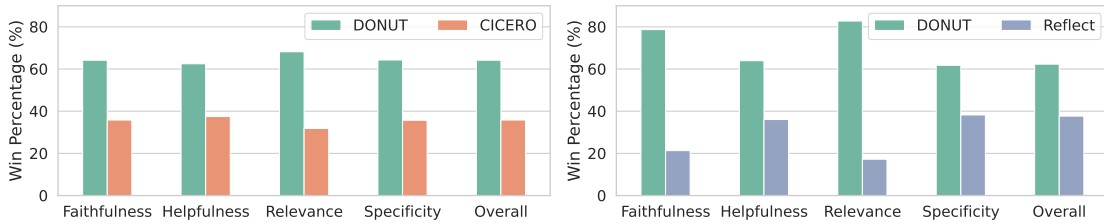

Figure 3: Results of head-to-head comparison between rationales from DONUT, commonsense annotation from CICERO (Ghosal et al., 2022a), and Reflect (Zhou et al., 2022a) via human judgment. The y-axis represents the win percentage against other datasets. The differences in all of the categories are statistically significant ($p < 0.05$).

ChatGPT, the annotation process takes 0.1 seconds per sample and costs 0.003 USD with a total of 1,200 USD. This significantly reduces the time and cost for data annotation.

**High quality.** We conduct a human evaluation via Amazon Mechanical Turk to compare the quality of CoT rationales from DONUT with Reflect-9K and CICERO. At each voting stage, three human judges are given two dialogues, one from DONUT and one from CICERO or Reflect-9K, and asked to choose a sample with better commonsense annotation based on five criteria: (1) faithfulness, (2) helpfulness, (3) relevance, (4) specificity, and (5) overall. To avoid potential bias from different annotation formats (*i.e.*, commonsense knowledge vs. rationales), we only use the last answer in the QA pairs from DONUT. Figure 3 presents human evaluation results on 100 randomly sampled dialogues. Judges deem commonsense inferences (*i.e.*, rationales) from DONUT superior in quality to the two human-authored ones across all aspects, validating the outstanding quality of the dataset.

### 4.2 Effects of Rationale Alignment

To investigate the effect of rationale alignment, we conduct additional human evaluations on rationales that have passed and failed each alignment filter via Amazon Mechanical Turk (AMT). From DONUT, we randomly sample 100 dialogues that contain two rationales, one that has passed the alignment filter (*i.e.*, pass) and another that has been filtered out (*i.e.*, fail). For each dialogue sample, human judges are given the two rationales and asked to choose the better one based on the same criteria used for quality assessment (§4.1).

Table 3 compares the win percentages between the two sets of rationales (*i.e.*, pass vs. fail) for each alignment filter. We observe that judges tend to find a rationale to be more consistent if it passes

| Win Percentage | R-to-Context | | | R-to-Response | | |
|---|---|---|---|---|---|---|
| | Pass | vs. | Fail | Pass | vs. | Fail |
| Consistency | **71%**[*] | | **29%**[*] | 59%[*] | | 41%[*] |
| Helpfulness | 53% | | 47% | **74%**[*] | | **26%**[*] |
| Specificity | 64%[*] | | 36%[*] | 63%[*] | | 37%[*] |
| Overall | 60%[*] | | 40%[*] | 61%[*] | | 39%[*] |

Table 3: Human evaluation results for head-to-head comparison between passed and filtered rationales from each filter (**R** = Rationale). Win percentages with [*] indicate significance ($p < 0.05$).

the rationale-to-context alignment filter. The same applies to the rationale-to-response filter, where judges tend to consider a rationale that passed the filter to be more helpful. These findings are in line with our intuition that aligning rationales with dialogue contexts and ground-truth responses improves consistency and helpfulness of the generated rationales, respectively.

## 5 Experiments

Our work builds upon previous efforts (Zhou et al., 2022b; Shen et al., 2022; Zhou et al., 2022a) to enrich dialogue models by injecting external commonsense knowledge. Hence, we conduct extensive experiments on response generation to examine how dialogue CoT reasoning from DOCTOR provides commonsense knowledge to assist dialogue agents in generating high-quality responses.

### 5.1 Datasets

**In-domain.** We evaluate DOCTOR on held-out test sets of the three datasets used for knowledge distillation: DailyDialog (Li et al., 2017), DREAM (Sun et al., 2019), and MuTual (Cui et al., 2020). These datasets contain open-domain dialogues that require commonsense knowledge to thoroughly understand dialogue contexts (Ghosal et al., 2022a). Note that DREAM and MuTual are

| Method | In-Domain | | | | | | | | | | | | Out-of-Domain | | | |
|---|---|---|---|---|---|---|---|---|---|---|---|---|---|---|---|---|
| | DailyDialog | | | | DREAM | | | | MuTual | | | | Reflect-9K | | | |
| | B-1 | B-2 | B-4 | R-L | B-1 | B-2 | B-4 | R-L | B-1 | B-2 | B-4 | R-L | B-1 | B-2 | B-4 | R-L |
| **Cosmo** [3B] | 20.04 | 8.12 | 2.38 | 13.69 | 20.98 | 8.35 | 2.28 | 14.03 | 17.63 | 6.00 | 1.30 | 12.03 | 14.97 | 4.24 | 0.70 | 11.8 |
| + COMET w/ ComFact | 19.54 | 7.82 | 2.25 | 14.05 | 20.90 | 8.26 | 2.17 | 14.09 | 17.56 | 6.24 | 1.37 | **12.41** | 15.80 | 4.60 | 0.87 | 12.09 |
| + DIALeCT | 19.63 | 8.05 | 2.36 | 14.31 | 20.69 | 8.18 | 2.17 | 13.82 | 18.01 | 6.35 | 1.31 | 12.23 | 15.54 | 4.55 | 0.85 | 11.82 |
| + Reflect | 19.44 | 7.62 | 2.11 | 13.58 | 18.23 | 7.10 | 1.91 | 13.30 | **18.57** | 6.13 | 1.31 | 12.01 | 15.33 | 4.20 | 0.71 | 12.05 |
| + DOCTOR (Ours) | **20.43** | **8.54** | **2.63** | **14.68** | **21.26** | **8.65** | **2.46** | 14.26 | 17.90 | **6.56** | **1.59** | 12.35 | **16.66** | **4.89** | **0.92** | **12.11** |
| ChatGPT [175B] | 17.25 | 7.18 | 2.11 | 14.72 | 18.90 | 7.81 | 2.29 | 14.86 | 17.92 | 7.03 | 1.79 | 14.83 | 17.28 | 5.29 | 1.12 | 12.77 |
| + COMET w/ ComFact | 18.24 | 7.50 | 2.19 | 14.56 | 20.09 | 8.06 | 2.33 | 14.44 | 19.32 | 7.74 | 2.18 | 15.46 | 17.38 | 5.30 | 1.03 | **13.28** |
| + DIALeCT | 16.61 | 6.49 | 1.82 | 13.55 | 18.00 | 6.85 | 1.88 | 13.16 | 19.15 | 7.67 | 1.85 | **15.55** | 17.48 | 5.29 | 1.09 | 12.96 |
| + Reflect | 17.47 | 6.98 | 2.05 | 13.80 | 19.02 | 7.35 | 2.10 | 13.37 | 18.14 | 6.87 | 1.91 | 14.27 | 18.24 | 5.46 | 1.15 | 12.54 |
| + Self-CoT | 18.16 | 7.24 | 2.22 | 12.62 | 18.88 | 7.13 | 2.01 | 12.17 | 19.97 | 7.32 | 1.82 | 13.71 | 14.53 | 4.28 | 0.83 | 11.56 |
| + DOCTOR (Ours) | **19.61** | **8.44** | **2.69** | **15.63** | **21.20** | **8.71** | **2.56** | **14.93** | **20.19** | **8.35** | **2.52** | 15.55 | **18.54** | **5.59** | **1.16** | 12.85 |

Table 4: Automatic evaluation results on DailyDialog, DREAM, and MuTual using BLEU (Papineni et al., 2002) and ROUGE (Lin, 2004). We use B-1/2/4, and R-L to denote BLEU-1/2/4, and ROUGE-L for simplicity.

designed for dialogue comprehension, but we adapt these datasets into response generation setting to fully leverage their high-quality conversations[9].

**Out-of-domain.** To assess the generalizability of DOCTOR, we consider Reflect-9K (Zhou et al., 2022a) as an additional dialogue dataset. Reflect-9K contains dialogues annotated with common knowledge between speakers. Note that we label this dialogue dataset as out-of-domain since it is an unseen dataset that has not been used to train DOCTOR, posing a challenge to its generalization.

## 5.2 Dialogue Agents

We consider two large-scale dialogue agents, Chat-GPT and Cosmo (Kim et al., 2022a). ChatGPT is an LLM with 175B parameters, trained to follow instructions. Cosmo is a general dialogue model trained with a million-scale dialogue corpus on top of T5 (Raffel et al., 2020; Lester et al., 2021). For our experiments, we use the 3B version of Cosmo[10].

Specifically, we augment both dialogue models with commonsense knowledge in a zero-shot manner to assess whether knowledge sources can be readily used to assist state-of-the-art dialogue models. To incorporate commonsense knowledge into dialogue agents, we simply prepend commonsense inference to the dialogue history as string concatenation (Zhou et al., 2022b; Kim et al., 2022b), where commonsense knowledge and dialogue history are separated using indicators (*e.g.*, <SEP>). We include the details on dialogue models in Appendix A.8 and our ChatGPT prompt in Table 13.

## 5.3 Baselines

To assess whether and how different knowledge sources affect the quality of generated responses, we compare DOCTOR with the following baselines. **(1) Without commonsense knowledge**: We first adopt the standard response generation baseline, where the dialogue agents predict responses conditioned on the dialogue history only. **(2) General-purpose commonsense knowledge model**: We then consider dialogue agents augmented with a general-purpose commonsense knowledge model COMET (Hwang et al., 2021). To align knowledge from COMET with dialogues, we implemented a retriever using ComFact (Gao et al., 2022) that retrieves relevant triplets to the dialogue context. **(3) Dialogue-focused commonsense knowledge model**: Finally, we construct task-specific baselines with knowledge models tailored for dialogue understanding. Specifically, we implement two knowledge models, DI-ALeCT (Shen et al., 2022) and Reflect (Zhou et al., 2022a), trained on dialogue datasets with qualified commonsense knowledge annotations. DIALeCT is trained on DailyDialog, DREAM, and MuTual, which are also used to train DOCTOR. Reflect, on the other hand, is trained on Reflect-9K which is tested as an out-of-domain dataset for DOCTOR. When tested on Reflect-9K, the model produces commonsense knowledge conditioned on oracle information, which is not given to DOCTOR. See Appendix A.6 for more details.

Note that both general-purpose and dialogue-focused knowledge models are **single-hop** approaches, as they are not designed to handle multi-hop reasoning in conversations.

---

[9]For MuTual, we retain the original dataset of dialogue context and ground truth response pairs to maintain the integrity of the original setup.

[10]https://huggingface.co/allenai/cosmo-xl

| Model | Natural | Consistent | Specific | Engaging |
|---|---|---|---|---|
| w/o CS | 33%* | 47% | 42%* | 48% |
| DOCTOR | **67%*** | **53%** | **58%*** | **52%** |
| ComFact | 32%* | 40%* | 44% | 49% |
| DOCTOR | **68%*** | **60%*** | **55%** | **51%** |
| DIALeCT | 18%* | 42%* | 43%* | 37%* |
| DOCTOR | **82%*** | **58%*** | **57%*** | **63%*** |
| Reflect | 33%* | 41%* | 48% | **53%** |
| DOCTOR | **67%*** | **59%*** | **52%** | 47% |
| Self-CoT | 33%* | 46% | 42%* | 43%* |
| DOCTOR | **67%*** | **54%** | **58%*** | **57%*** |

Table 5: Human evaluation results of responses from ChatGPT on DailyDialog when paired with DOCTOR vs. baseline models. "w/o CS" denotes direct response generation without commonsense knowledge. Win percentages with * indicate significance ($p < 0.05$).

| Model | Consistent | Helpful | Specific | Overall |
|---|---|---|---|---|
| ComFact | 33%* | 42%* | 38% | 48 % |
| DOCTOR | **76%*** | **58%*** | **62%*** | **52%** |
| DIALeCT | 29% * | 39%* | 48% | 44 %* |
| DOCTOR | **71%*** | **61%*** | **52%** | **56%*** |
| Reflect | 18%* | 29%* | 28% | 33% |
| DOCTOR | **82%*** | **71%*** | **72%*** | **67*%** |
| Self-CoT | 27%* | 44% | **52%** | 46% |
| DOCTOR | **73%*** | **56%*** | 48% | **54%** |

Table 6: Human evaluation results of the quality of commonsense knowledge from DOCTOR vs. baseline knowledge modes. Win percentages with * indicate significance ($p < 0.05$).

## 5.4 Main Results

We report the results of the automatic evaluation in Table 4 and human evaluation via Amazon Mechanical Turk in Table 5. For examples of generated responses, see Appendix E.

**Helpfulness of dialogue CoT rationales.** On both automatic (Table 4) and human evaluations (Table 5), we observe that integrating dialogue CoT into dialogue models improves their performance over the vanilla dialogue models without commonsense. Table 5 shows that responses conditioned on dialogue CoT are particularly more natural and specific than those from vanilla ChatGPT. We also observe from Table 4 that DOCTOR generates helpful rationales for the dialogue models on Reflect-9K, which is not used to train DOCTOR. While these dialogue models are trained on significantly large-scale datasets, they still benefit from DOCTOR in capturing implicit information in conversations.

**Comparison with single-hop approaches.** Table 4 compares the performance of dialogue models paired with the single-hop baselines, *i.e.* general-purpose and dialogue-focused commonsense knowledge models. We find that augmenting dialogue models with baseline knowledge models show only a slight improvement and sometimes even a subtle decrease in performance compared to the vanilla model. These results suggest that the baselines struggle to produce correct reasoning

with limited knowledge of implicit contexts.

Overall, CoT rationales from DOCTOR lead to a larger improvement over the baselines. In particular, we find that dialogue models augmented with DOCTOR outperform the models paired with Reflect, which serves as an oracle in the unseen benchmark Reflect-9K. Furthermore, human evaluation results in Table 5 show that responses grounded to dialogue CoT rationales tend to be more natural and helpful compared those grounded to baseline knowledge models such as DIALeCT, which is trained using the same corpora.

**Comparison with self-generated CoT.** To examine the validity of LLMs as dialogue CoT reasoners, we compare the performance of dialogue agents augmented with CoT rationales from DOCTOR and the teacher LLM (*i.e.* ChatGPT). Specifically, we instruct ChatGPT with the same demonstrations used in DONUT construction to generate dialogue CoT rationales and predict the next response conditioned on them. Surprisingly, we observe in Table 4 that augmenting dialogue models with CoT rationales from ChatGPT, denoted as Self-CoT, does not lead to better response quality over DOCTOR. This result shows that LLMs do not reliably produce helpful rationales for response generation, suggesting the need for alignment filter to control the quality of the rationales.

## 5.5 Analysis

**Better knowledge leads to better responses.** To better understand the effect of knowledge on response generation, we conduct a human evaluation

| Training | B-1 | B-2 | B-4 | R-L |
|---|---|---|---|---|
| DONUT (full) | **19.61** | **8.44** | **2.69** | **15.63** |
| DONUT (answer-only) | 18.45 | 7.85 | 2.49 | 15.28 |

Table 7: Results of ablation on generating question using the response quality of ChatGPT on DailyDialog.

| Filtering | Natural | Consistent | Specific | Engaging |
|---|---|---|---|---|
| w/o R-to-C | 30%* | 42%* | **53%** | 44%* |
| w/ R-to-C | **70%** * | **58%** * | 47% | **56%** * |
| w/o R-to-R | 36%* | 43%* | 49% | 44%* |
| w/ R-to-R | **67%** * | **57%** * | **51%** | **56%** * |

Table 8: Human evaluation results of responses from ChatGPT on DailyDialog when paired with the models trained with different alignment filter settings. "R-to-C" and "R-to-R" denote rationale-to-context/response alignment filters respectively. Win percentages with * indicate significance ($p < 0.05$).

of the quality of knowledge. We randomly sample 100 knowledge inferences on the test set of Daily-Dialog and ask three different human judges for each sample to compare the knowledge generated by DOCTOR and the baseline model on the same aspects used in §4.1. For the evaluation details, see Appendix C. The results are shown in Table 6. While the baselines produce knowledge relevant to the dialogue contexts, the knowledge lacks consistency and is usually unhelpful in predicting the responses. Since DOCTOR generates CoT rationales by grounding on implicit evidence aggregated over multiple turns, knowledge from DOCTOR is far superior in terms of specificity and helpfulness, which in turn leads to better response quality.

**Iterative QA helps response generation.** To analyze the role of questions, we conduct an ablation on the generation of questions. We train an ablated model under the same setting as DOCTOR to generate only answers. Specifically, we explicitly remove questions from the rationale annotations in DONUT and train the model with the reconstructed data. We use ChatGPT for the dialogue model and compare the response quality on DailyDialog. The results are shown in Table 7. Without questions, the response quality drops significantly, suggesting the importance of the questions in rationalization. We posit that the role of questions in guiding the answers is crucial, as the answers are poorly aligned with the dialogues without guidance.

**Applying filters improves response quality.** To analyze the impact of the alignment filters, we train two reasoning models with only passed and filtered rationales respectively for each filter. For fair comparisons, in training, we use the same amount of rationale labels that are aligned with the same context. We show the results in Table 8. The performance of the dialogue model drops significantly when the reasoning model is trained without the rationale-to-context alignment filter, suggesting the importance of the alignment between rationales and contexts. Also, when the model is trained with the rationales not aligned with the responses, the quality of the

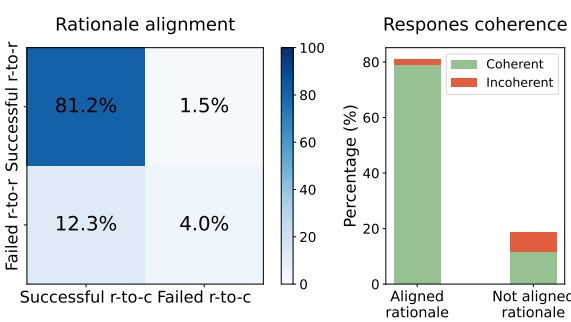

Figure 4: Results of qualitative analysis. **Left** shows the proportions of rationales that are aligned with context (r-to-c) or response (r-to-r). **Right** shows the percentage of coherent responses by rationale alignment.

response decreases, as the generated rationales are not helpful in predicting the next responses.

To gain a deeper understanding on the effect of well-aligned rationales on the response quality, we perform an in-depth analysis on the 600 evaluation examples randomly sampled from DailyDialog. For each sample, we present human annotators from AMT with the dialogue context, rationales, reference response and predicted response and ask them to answer (1) whether the knowledge is aligned with the dialogue context, (2) whether the knowledge is aligned with the reference response, and (3) whether the predicted response is coherent with the dialogue context.

In Figure 4, we find that 81.2% of the annotated rationales are considered to be aligned with both dialogue contexts and gold responses, suggesting that DOCTOR trained using filtered data from DONUT learns to generate rationales aligned with both the dialogue contexts and the responses. We also observe that only 2.1% out of 81.1% of samples with aligned rationales are deemed incoherent, indicating that the generated response tends to be coherent

if the provided rationale is well aligned. We provide an error analysis on the generated rationales and responses in Appendix B.

# 6 Related Work

**Commonsense-aware dialogue models.** Recent studies incorporate commonsense knowledge into dialogue models to facilitate engaging interactions between humans. These approaches leverage knowledge from a general-purpose knowledge model (Zhou et al., 2022b; Wu et al., 2022; Liu et al., 2022b; Li et al., 2023) or a dialogue-focused knowledge model trained with human-annotated dataset (Ghosal et al., 2022a; Gao et al., 2022). On the other hand, we focus on building a knowledge model for multi-hop commonsense reasoning in dialogues, where desired knowledge is induced from implicit evidence scattered in dialogue contexts.

**Chain-of-Thought reasoning.** LLMs have shown an emergent capability in reasoning by eliciting rationales as explanations via CoT prompting (Wei et al., 2022; Wang et al., 2023b; Zhou et al., 2023). Despite their promising ability, we find that applying CoT prompting in dialogues is a non-trivial challenge even for LLMs.

Meanwhile, recent work proposes distillation frameworks for transferring the reasoning ability of LLMs to small language models (Wang et al., 2023a; Kang et al., 2023). However, these approaches focus on generating rationales for answering factoid questions and are suboptimal for commonsense reasoning in dialogues. This motivates the need to selectively transfer CoT rationales from LLMs in conversations.

# 7 Conclusion

Commonsense reasoning in conversations involves multi-hop reasoning, which poses challenges even for LLMs. To address this, we present a dialogue chain-of-thought distillation framework that selectively annotates high-quality rationales using LLMs. Our contributions are as follows: (1) With our framework, we collect DONUT, a large-scale dataset for dialogue CoT reasoning. (2) We present DOCTOR, a dialogue chain-of-thought reasoner trained on DONUT. (3) Through extensive experiments, we show the efficacy of DOCTOR, especially in the human evaluation, where 67% of the responses generated using DOCTOR are preferred over the responses using knowledge from LLMs.

# Limitations

We test DOCTOR on 4 diverse dialogue datasets, but our experiments are limited to open-domain and dyadic dialogues. Further study can apply DOCTOR to task-oriented or multi-party dialouges. Further studies could adjust this variable dynamically, potentially allowing for deeper levels of dialogue reasoning. The rationale annotations in DONUT are fully machine-generated. Caution must be exercised when training models with them.

# Ethical Considerations

Texts generated by a large language model can contain harmful, biased, or offensive content. However, we argue that this risk is mostly mitigated in our work, as we focused on the knowledge within widely-used popular dialogue datasets. The four source datasets: DailyDialog, MuTual, DREAM, and SODA are either high-quality datasets authored by humans or examined via safety filtering mechanisms (both models and web-based API) targeting crimes, violence, hate, sexuality, etc. We also examine the generated rationales and manually eliminate toxic and offensive uses of language. We guarantee fair compensation for judges we hire on Amazon Mechanical Turk. We ensure an effective pay rate higher than $15 per hour based on the estimated time required to complete the tasks. The presented DONUT dataset does not contain personal data or information that provides clues for the identification of any individual or group.

# Acknowledgements

This work was supported by Institute of Information & Communications Technology Planning & Evaluation (IITP) grant funded by the Korean government (MSIT) (No. 2020-0-01361, Artificial Intelligence Graduate School Program (Yonsei University)) and (No.2021-0-02068, Artificial Intelligence Innovation Hub) and (No.2022-0-00077, AI Technology Development for Commonsense Extraction, Reasoning, and Inference from Heterogeneous Data). Jinyoung Yeo, Dongha Lee, and Youngjae Yu are co-corresponding authors.

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

## A  Experimental Details

### A.1  Rationale Candidate Generation

We prompt ChatGPT to annotate CoT rationales for a given dialogue context (*i.e.*, history) and the ground-truth response. We set temperature to 0.5, and max tokens to 300. The generation is led by subquestions based on several selected types of commonsense relations. Following West et al. (2022) and Gao et al. (2022), we carefully choose 11 relation types which are crucial in conversations (Zhong et al., 2022; Ghosal et al., 2022b; Zhou et al., 2022a) from ATOMIC (Hwang et al., 2021) as presented in Table 9. The prompt used for rationale generation is shown in Table 12.

| Relation type | Example question |
|---|---|
| xIntent | What is the plan that *speaker* and *listener* have made? |
| xNeed | What does *speaker* need to do to pass the final exam? |
| xReact | How might *speaker* react to the breaking news from *listener*? |
| xWant | What does *speaker* want to know from *listener*? |
| xAttr | What is *speaker*'s role? |
| oEffect | What is the result of *listener*'s inquiry about George Hatton? |
| oReact | What will *listener* react after confirming the meeting time and place? |
| oWant | What does *listener* want to convey to *speaker* about the prices? |
| isAfter | What might *listener* request from *speaker* after the agreement? |
| isBefore | What happened before *speaker*'s first trip abroad? |
| Causes | What causes *listener* to be concerned about being late? |

Table 9: Relation types and example questions.

### A.2  Ablation on number of reasoning steps

To better understand the effect of $k$ on the quality of rationale, we conduct human evaluation using 100 random dialogue samples from DailyDialog, DREAM, and MuTual. For each dialogue, we prompt ChatGPT to generate five CoT rationales with $k = \{1, 2, 3, 4, 5\}$, respectively, as we do in §3.1. Using the same criteria from Table 3, we ask 3 different workers from Amazon Mechanical Turk to evaluate the quality of the rationale from each dialogue. The results are shown in Table 10.

| $k$ | Consistency | Helpfulness | Specificity | Overall |
|---|---|---|---|---|
| 1 | 78.1 | 77.9 | 83.4 | 79.9 |
| 2 | 87.5 | 78.4 | 81.1 | 80.7 |
| 3 | 91.2 | 81.4 | 86.9 | 87.1 |
| 4 | 88.5 | 78.1 | 88.0 | 83.6 |
| 5 | 86.9 | 76.5 | 83.2 | 82.6 |

Table 10: Human evaluation results on rationales with different $k$.

The workers prefer the rationales with $k = 3$ most in terms of consistency, helpfulness, and overall. The Krippendorff alpha (0.82, 0.58, 0.74, 0.71) scores show a moderate agreement among the raters.

### A.3  Rationale-to-context Alignment Filter

**Data collection.** To collect training data for the rationale-to-context alignment filter, we randomly sample 6K dialogues from SODA (Kim et al., 2022a) that do not overlap with those used as source dialogues for our DONUT. For each dialogue (context and ground-truth response), we first remove all utterances in the context except for the last one to obtain an incomplete context, and prompt an LLM to generate rationales based on the original and incomplete contexts respectively. For the rationales generated with the original contexts, we manually select one rationale that is well-aligned with the context. We, therefore, acquire a rationale that is grounded on the whole context along with a *counterfactual* rationale, which ought to be inconsistent with the dialogue as it merely considers the last utterance.

This results in 6K dialogues aligned with their rationales and counterfactual rationales. We duplicate the dialogues and align them with either type of rationale for the binary classification of rationale-to-context alignment. The data (12K) is split into training (10K), validation (1K), and test set (1K) without the overlap of dialogues across them.

**Training.** To train this alignment filter, we use the aforementioned data to finetune the RoBERTa-large (Liu et al., 2019) model for 3 epochs, with a batch size of 40 and a learning rate of 1e-5[11]. The training is run on one NVIDIA RTX A5000 GPU. The classification performance of this filter on the test set in terms of accuracy is 93.38%.

---

[11]https://huggingface.co/roberta-large

### A.4 Rationale-to-response Alignment Filter

Following Liu et al. (2022a), we use the perplexity of the ground-truth response for calculating $P_\theta(u_t|\cdot)$. For efficiency, we implement the filter with 4-bit quantization. The filtering process on whole rationale candidates with rationale-to-response alignment filter takes about 8 GPU hours with 8 NVIDIA RTX A5000 GPUs.

### A.5 DOCTOR

We train DOCTOR with DONUT for 5 epochs using a constant learning rate of 5e-4 on 8 NVIDIA RTX A5000 GPUs. We use a batch size of 8, and the whole training process takes about 12 GPU hours. For efficiency, we adopt 16-bit quantization for training.

### A.6 Baseline Knowledge Models

**ComFact.** Following the setting of Gao et al. (2022), we use COMET and the same relation types (*i.e.*, xReact, xIntent, xNeed, xEffect, and xWant). We use COMET-BART to implement COMET[12]. We use the same decoding strategy used by Gao et al. (2022) to generate inference with COMET. We implement the knowledge retriever using the source code on the official GitHub repository[13] using DeBERTa-large. Among the inferences generated by COMET, we apply the retriever and choose the inferences predicted as relevant to the dialogue context.

**DIALeCT.** We take the CICERO v1 dataset from Ghosal et al. (2022a) and convert the data format following DONUT. We then fine-tuning the OPT-1.3B (Zhang et al., 2022) with the converted data[14]. The training details are identical to DOCTOR. Taking the name, DIALeCT, from Shen et al. (2022), we re-implemented the model to generate common-sense inference with CICERO v1. When we generate inferences with DIALeCT, we use all the question types defined in CICERO v1 (*i.e.*, "What is or could be the cause of target?", "What subsequent event happens or could happen following the target?", "What is or could be the prerequisite of target?", "What is the possible emotional reaction of the listener in response to target?", and "What is or could be the motivation of target?") and we set the target of inference as the last utterance in the dialogue history. We concatenate all generated

inferences with the newline characters and prepend it to the dialogue history for response generation.

**Reflect.** We finetune the OPT-1.3B with the Reflect-9K dataset (Zhou et al., 2022a). It is a human-authored dialogue dataset with aligned inference sentences that approximate common beliefs between interlocutors. Specifically, we concatenate the annotated question and the dialogue history as inputs and train a knowledge model to predict the paired inference. The training details are identical to DOCTOR. When inference, we generate all types of inference dimensions defined in Reflect-9K (*i.e.*, "How would you describe Speaker?", "What might have happened before?", "What might happen after?", "What is Speaker feeling now?", and "What Responder feeling now?"). For response generation, we concatenate inferences generated by this knowledge model using the newline characters and prepend to the dialogue history.

### A.7 Self Chain-of-Thought

We prompt ChatGPT to generate CoT rationales and predict the target response based on the dialogue context and self-generated rationales. We use the same demonstrations as the prompt used in our framework and instruct ChatGPT to generate a dialogue CoT rationale and the following response.

### A.8 Dialogue Agents for Response Generation

**Cosmo.** Cosmo is a dialogue model trained with a million-scale dialogue corpus on top of T5 (Lester et al., 2021). We use the 3B version of Cosmo[15]. To generate the response from Cosmo, we use greedy decoding, and 4-bit quantization for efficiency. When the length of the input context exceeds 512 tokens, we truncate the sequence from left to ensure the last utterance is not removed from the input. We use a special token <SEP> to separate commonsense knowledge and dialogue history. Cosmo is trained to generate responses conditioned on a narrative expanded from commonsense sentences, which are separated by a dialogue context using the <SEP>.

**ChatGPT.** ChatGPT is an LLM with 175B parameters based on InstructGPT (Ouyang et al., 2022)[16]. ChatGPT is trained to follow instructions given by users and return requested information in

---

[12]https://github.com/allenai/comet-atomic-2020
[13]https://github.com/silin159/comfact
[14]https://huggingface.co/facebook/opt-1.3b

[15]https://huggingface.co/allenai/cosmo-xl
[16]https://openai.com/blog/chatgpt

a conversational manner. We use langchain[17] to send API calls to OpenAI API.

We prompt ChatGPT to predict the next response based on the dialogue context (*i.e.*, history) and the augmented knowledge The prompt used for response generation is in Table 13. We append speaker tags (*e.g.*, "A:") to enforce the model to predict the next response and not confuse which speaker takes the next turn.

## B  Error Analysis

For error analysis, we collect 600 random samples from the test set of DailyDialog. We present workers from AMT with the dialogue context, rationales, the reference response and the predicted response and ask them to evaluate generated rationales and predicted responses by answering the following yes-no questions:

- Do you agree that knowledge is well aligned with the dialogue context?

- Do you agree that knowledge is well aligned with the reference response?

- Do you agree that the predicted response is coherent with the dialogue context?

Each sample is evaluated by 3 different workers to reduce variance and improve the reliability of the evaluation. To collect error cases, we manually inspect samples where at least 2 workers disagree with the statement in each question. Refer to Figure 4 for the statistics of our evaluation.

Among all test examples, DOCTOR generates rationales that are not aligned with the dialogue contexts for only 5.5% of the cases. We observe two major error types behind such misalignment with the dialogue contexts: (1) for 49% of the error cases, DOCTOR struggles to follow the complex dialogue flow and does not answer the questions correctly, failing to aggregate enough evidence even for the correct subquestions; (2) for 38% of the error cases, DOCTOR concludes the rationale with a statement that cannot be induced from the dialogue context, mostly because it is either too short or not specific enough to contain necessary evidence for coherent reasoning.

We also find only 16.3% of the test samples where DOCTOR generates rationales that are not aligned with the reference responses. The two major reasons behind the misalignment between the

---

[17]https://github.com/hwchase17/langchain

rationales and the responses are as follows: (1) 33% of the error cases contain plausible rationales from the dialogue context that lead to responses different from the reference since the openness of dialogue allows for multiple possible responses for a single dialogue context. (2) for 31% of the error cases, DOCTOR generates sophisticated rationales to describe its reasoning even in scenarios where simple conversations are enough. *e.g.*, daily greetings.

As discussed in Section 5.5, we observe in Figure 4 that few samples with aligned rationales lead to incoherent responses from the dialogue model. One possible reason behind these few failure cases is that rationales from DOCTOR might be too complex and lengthy due to the complex nature of dialogue. In such cases, chat LLMs sometimes fail to fully reflect the rationales in their responses, leading to incoherent responses.

## C  Details for the Human Evaluation

### C.1  Annotated Knowledge Quality

We outsource a human evaluation comparing our DONUT and human-authored datasets on Amazon Mechanical Turk (AMT). We show the interface for the evaluation in Figure 7. We ask the human judges to compare the annotated knowledge from each dataset based on the following five criteria:

- Faithfulness: Which knowledge statement is less contradictory to its aligned dialogue context and target response?

- Helpfulness: Which knowledge statement is more helpful in predicting the target response?

- Relevance: Which knowledge statement is more relevant to its aligned dialogue context?

- Specificity: Which knowledge statement is more specific and focused on the target response?

- Overall: Overall, which knowledge statement is more useful and valuable?

At each voting stage, human judges are given two dialogues with aligned commonsense inferences and asked to select a better knowledge statement according to the above criteria. We show answers in our rationales without the prefix (*i.e.*, "Subquestion:") to match the format.

## C.2 Knowledge Inference Quality

We conduct human evaluation of the quality of inferred knowledge via AMT. The interface for evaluation is shown in Figure 8. We ask human judges to compare the knowledge from DOCTOR and the baseline knowledge models. We focus on these four criteria:

- Consistency: Which inference sentence is more consistent with the dialogue context?

- Specificity: Which inference sentence is more specific and focused on the target response?

- Helpfulness: Which inference sentence is more helpful in predicting the target response?

- Overall: Overall, which inference sentence is more useful and valuable?

## C.3 Response Quality

We also outsource a human evaluation for comparing the responses from ChatGPT when paired with DOCTOR and the baseline knowledge models on AMT. We ask human judges to compare the responses based on these four criteria following Kim et al. (2022a) and Zhou et al. (2022a):

- Naturalness: Which response is more natural (human-like)?

- Consistency: Which response is more consistent (well aligned) with the dialogue context?

- Specificity: Which response is more specific?

- Engagingness: Which response is more engaging?

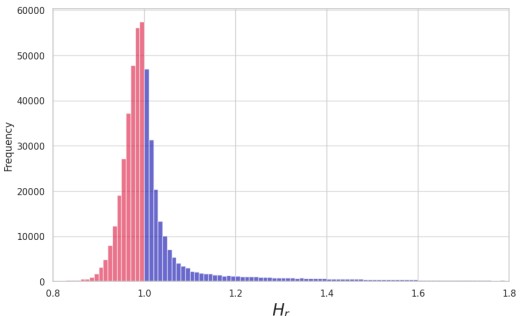

Figure 5: Distribution of the helpfulness ratio.

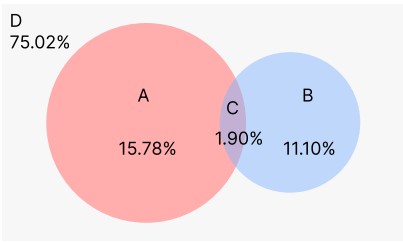

**A.** Filtered out by the rationale-to-context filter.
**B.** Filtered out by the rationale-to-response filter.
**C.** Filtered out by both filters. **D.** Pass both filters.
**Filtered rationales in total** = A + B - C = 24.98%

Figure 6: Percentage of filtered rationales.

| Top | Step 1 | | Step 2 | | Step 3 | |
|---|---|---|---|---|---|---|
| 1 | xAttr | 39.7% | oReact | 54.9% | xWant | 24.1% |
| 2 | xIntent | 28.4% | xAttr | 13.6% | oWant | 18.5% |
| 3 | xReact | 8.9% | xIntent | 5.5% | oReact | 17.4% |
| 4 | oReact | 8.8% | xIntent | 5.5% | xIntent | 11.6% |
| 5 | xWant | 5.8% | oWant | 5.0% | oEffect | 6.8% |

Table 11: Top 5 relation types at each generation step.

# D Statistical Study on Generated Rationales and Alignment Filters

## D.1 Distribution of the Helpfulness Ratio

In Figure 5, we provide the distribution of the helpfulness ratio $H_r$ from the rationale-to-response alignment filter:

$$H_r = \frac{P_\theta(u_t|Z, U_{<t})}{P_\theta(u_t|U_{<t})} \quad (5)$$

The mean of all $H_r$ is 1.076 with a standard deviation of 0.496.

## D.2 Why Two Alignment Filters?

Figure 6 illustrates the percentage of rationales filtered out by the two alignment filters targeting rationale-to-context and rationale-to-response alignment, respectively. The two filters together filter out 24.98% of the rationales in total, while the rationales being simultaneously filtered out by both filters only accounted for 1.9% of all the samples. This manifests the necessity of implementing filters that target different types of alignment. We include examples that have gone through the two filters in Table 14 to 16.

## D.3 Relation Types in Rationales

As a formal framework for CoT reasoning, we investigate if questions in the QA sequence evolve from more generic types of commonsense relation to more specific ones. Table 11 is the distribution

of commonsense relation types in each generation step. We present the top five most used relation types. In the first step, a large portion (39.7%) of questions focused on acquiring information about the participants of the dialogue (xAttr), facilitating better alignment between the rationales and context. In the last step, the most questions are related to the intention of the speaker (xWant). By inferring the communicative intent of speakers, the rationale could be more helpful in predicting the next response.

## E   Examples of Response Generation with Different Knowledge Sources

We show some examples of response generation using ChatGPT with different knowledge sources in Table 17, Table 18, and Table 19.

We are studying meaningful **evaluation metrics** for the **qualities** of knowledge. The knowledge is aquired by answering consequent questions.

Specifically, you'll be given a piece of dialog, response and **two** knowledge, and you'll be asked to **compare which knowledge is better** in terms of specific aspects, **specify which aspect was most important** for judging, and **write down your rationales in free-text**.

---

*Guidelines:*
1. **[Q1~5] First, choose which knowledge is better regarding the given aspect.** There are four choices: `Definitely A/B` and `Slightly A/B` .
   - Please trust your instincts and choose `Definitely` if you would feel more confident giving one response, versus the other one.
   - Try to focus on quality over quantity. **Contentful/high-quality** knowledge doesn't need to be lengthy.
2. **[Q6] Second, choose which aspect influenced you the most when judging the overall quality.**
   - If some factor other than the ones in Question 1~5 had the biggest influence, please select "Other" and specify.
3. **[Q7] Third, please describe in detail your option for the questions.**
   - It would be helpful to describe both *reasons you like the better knowledge* **and** *reasons why you did not like the other knowledge.*
   - Please be specific and detailed in your rationale.

| *Dialog Context* | *Dialog Context* |
|---|---|
| ${context1} | ${context2} |

| *Response* | *Response* |
|---|---|
| ${response1} | ${response2} |

| *Knowledge A* | *Knowledge B* |
|---|---|
| ${knowledge1} | ${knowledge2} |

**Question 1. Which knowledge is more faithful (based on the dialog and the response)?**
     ● Definitely A   ● Slightly A   ● Slightly B   ● Definitely B

**Question 2. Which knowledge is more relevant to the dialogue context?**
     ● Definitely A   ● Slightly A   ● Slightly B   ● Definitely B

**Question 3. Which knowledge is more specific?**
     ● Definitely A   ● Slightly A   ● Slightly B   ● Definitely B

**Question 4. Which knowledge is more helpful in predicting the response?**
     ● Definitely A   ● Slightly A   ● Slightly B   ● Definitely B

**Question 5. Which knowledge do you like more overall?**
     ● Definitely A   ● Slightly A   ● Slightly B   ● Definitely B

**Question 6. Which aspect affected you the most when judging the overall quality?**
     ○ Faithfulness   ○ Relevance   ○ Specificity   ○ Helpfulness   ○ Other: __________

Figure 7: Interface for human evaluation on annotated knowledge quality.

We are studying meaningful **evaluation metrics** for the **qualities** of knowledge. The knowledge is aquired by answering consequent questions.

Specifically, you'll be given a piece of dialog, response and **two** knowledge, and you'll be asked to **compare which knowledge is better** in terms of specific aspects, **specify which aspect was most important** for judging, and **write down your rationales in free-text**.

*Guidelines:*
1. **[Q1~5] First, choose which knowledge is better regarding the given aspect.** There are four choices: **Definitely A/B** and **Slightly A/B** .
   ○ Please trust your instincts and choose **Definitely** if you would feel more confident giving one response, versus the other one.
   ○ Try to focus on quality over quantity. **Contentful/high-quality** knowledge doesn't need to be lengthy.
2. **[Q6] Second, choose which aspect influenced you the most when judging the overall quality.**
   ○ If some factor other than the ones in Question 1~5 had the biggest influence, please select "Other" and specify.
3. **[Q7] Third, please describe in detail your option for the questions.**
   ○ It would be helpful to describe both *reasons you like* the better knowledge **and** *reasons why you did not like* the other knowledge.
   ○ Please be specific and detailed in your rationale.

| ***Dialog Context*** |
| :---: |
| ${context} |

| ***Response*** |
| :---: |
| ${response} |

| ***Knowledge A*** | ***Knowledge B*** |
| :--- | :--- |
| ${knowledgea} | ${knowledgeb} |

**Question 1. Which knowledge is more faithful (based on the dialog and the response)?**
○ Definitely A  ○ Slightly A  ○ Slightly B  ○ Definitely B

**Question 2. Which knowledge is more consistent (well aligned) with the dialogue context?**
○ Definitely A  ○ Slightly A  ○ Slightly B  ○ Definitely B

**Question 3. Which knowledge is more specific?**
○ Definitely A  ○ Slightly A  ○ Slightly B  ○ Definitely B

**Question 4. Which knowledge is more helpful in predicting the response?**
○ Definitely A  ○ Slightly A  ○ Slightly B  ○ Definitely B

**Question 5. Which knowledge do you like more overall?**
○ Definitely A  ○ Slightly A  ○ Slightly B  ○ Definitely B

**Question 6. Which aspect affected you the most when judging the overall quality?**
○ Consistency  ○ Specificity  ○ Helpfulness  ○ Overall  ○ Other:

Figure 8: Interface for human evaluation on knowledge inference quality.

We are studying meaningful **evaluation metrics** for the **qualities** of response. The response is aquired by answering consequent questions.

Specifically, you'll be given a piece of dialog history, and response, and you'll be asked to **compare which response is better in terms of specific aspects**, **specify which aspect was most important** for judging, and **write down your rationales in free-text**.

---

*Guidelines:*
1. **[Q1~4] First, choose which response is better regarding the given aspect.** There are four choices: **Definitely A/B** and **Slightly A/B** .
   - Please trust your instincts and choose **Definitely** if you would feel more confident giving one response, versus the other one.
   - Try to focus on quality over quantity. **Contentful/high-quality** response doesn't need to be lengthy.
2. **[Q5] Second, choose which aspect influenced you the most when judging the overall quality.**
   - If some factor other than the ones in Question 1~4 had the biggest influence, please select "Other" and specify.
3. **[Q6] Third, please describe in detail your option for the questions.**
   - It would be interesting to describe both *reasons you like the better response* **and** *reasons why you did not like the other response*.
   - Please be specific and detailed in your rationale.

---

**Dialog Context**

${context}

---

**Response A**

${responsea}

**Response B**

${responseb}

---

**Question 1. Which response is more natural (human-like)?**

○ Definitely A    ○ Slightly A    ○ Slightly B    ○ Definitely B

---

**Question 2. Which response is more engaging?**

○ Definitely A    ○ Slightly A    ○ Slightly B    ○ Definitely B

---

**Question 3. Which response is more consistent (well aligned) with the dialogue context?**

○ Definitely A    ○ Slightly A    ○ Slightly B    ○ Definitely B

---

**Question 4. Which response is more specific?**

○ Definitely A    ○ Slightly A    ○ Slightly B    ○ Definitely B

---

**Question 5. Which aspect affected you the most when judging the overall quality?**

○ Natualness    ○ Engagingness    ○ Consistency    ○ Specificity    ○ Other: __________

Figure 9: Interface for human evaluation on response quality.

| Prompt |
| --- |

Generate rationales for generating the target utterance ("Target:"). The rationale consists of 3-hop subquestion-subanswer pairs.

Each question should contain a commonsense relation in [oEffect, oReact, oWant, xAttr, xIntent, xNeed, xReact, xWant, isAfter, isBefore, Causes]. These rationales should be the crucial cue for generating the target utterance, but you should not include the target utterance and also pretend you don't know the target utterance.

Subquestion 3 and Subanswer 3 should be about guessing the target utterance, so Subanswer 3 should be closely related to the target utterance but don't mention it directly.

If you think generating the target utterance doesn't need commonsense, then generate None for the rationale.

- Example 1 -
A: The mosquitoes are biting me!
B: Me too, I can't stop scratching. They are everywhere! Sneaky little jerks.
A: Do you have any bug spray?
B: No, I forgot to buy some.
A: Then we'll have to put up with it.
B: We can cover ourselves with beer! That way if they bite us, they'll get drunk and fall asleep.
A: That's without a doubt, the best idea you've had! Let's do it!

Ground-truth Response:
B: Run! They are thirsty for more!

Rationale:
Subquestion 1: What is the intent of Person B when suggesting the use of beer to ward off mosquitos? (xIntent)
Subanswer 1: Person B's intention is to make the mosquitos 'drunk' and cause them to fall asleep, reducing the amount of bites.
Subquestion 2: What is Person A's reaction to Person B's unique idea to use beer? (xReact)
Subanswer 2: Person A finds the idea amusing and agreeable, and shows enthusiasm in trying it out.
Subquestion 3: What might be the effect on the mosquitos after Person A and B use beer to ward them off? (oEffect)
Subanswer 3: Unexpectedly, the mosquitos might be attracted to the beer, causing them to swarm more intensively, creating the need for Person B to warn Person A about the increased mosquito activity.

- Example 2 -
A: Did you check the internet for next week's weather forecast ?
B: I sure did. You're in luck! It's supposed to snow all week in the mountains!
A: Yes! Somebody up there loves me! I knew it wasn't too late for snow.
B: It is kind of strange though, to have snow in April, and so much of it.
A: There have been so many dry winters lately that it's about time, don't you think?
B: When you put it that way, I guess the skies can't hold out on us forever.
A: This will be the best ski trip I've ever taken. I can't wait to use my new board.

Ground-truth Response:
B: It will be the best as long as the weatherman is right.

Rationale:
Subquestion 1: What is Person A's reaction to the snow forecast? (xReact)
Subanswer 1: Person A is excited about the snow forecast because he plans to use his new board on a ski trip.
Subquestion 2: What does Person B think about the reliability of weather forecasts? (xAttr)
Subanswer 2: Person B might believe that weather forecasts are not always accurate, given the unpredictability of weather patterns.
Subquestion 3: What might Person B want to communicate to Person A, given Person A's excitement and the uncertainty of weather forecasts? (oWant)
Subanswer 3: Person B might want to remind Person A that his ski trip being the best is contingent on the accuracy of the weather forecast.

- Example 3 -
A: ......
...

Table 12: **The Prompt for Generating Rationales.** We prompt ChatGPT to generate rationales in a five-shot setting (Example 3, 4, and 5 are omitted in this table). We concatenate the source dialogue at the end of the prompt.

| Prompt |
| --- |
| Generate the most plausible next response considering the dialogue history. You can refer to the rationale, but you should ignore the rationale if it misleads the next response. Do not try to put too much information in the next response. You should follow the style of the history.

Rationale:
{rationale}
History:
{history}
Next Response:
{name_tag} |

Table 13: **An Example Prompt for Response Generation.** We prompt ChatGPT to generate a response based on the dialogue context (history) and its aligned rationales in a zero-shot setting.

| Context | |
| --- | --- |
| A: How may I help you? | |
| B: I would like to return an item. | |
| A: What are you returning? | |
| B: I want to return this cellphone. | |
| A: Is there a problem? | |
| B: It's broken. | |
| **Response** | |
| A: What exactly is wrong with it? | |
| **Probability from Rationale-to-context Alignment Filter** | 0.99 |
| **Passed Rationale** | |
| Subquestion 1: Why did Person B come to the store? (xIntent) | |
| Subanswer 1: Person B wants to return a cellphone. | |
| Subquestion 2: What is the reason for Person B to return the cellphone? (xAttr) | |
| Subanswer 2: The cellphone is broken. | |
| Subquestion 3: What information does Person A need to process the return? (xNeed) | |
| Subanswer 3: Person A needs to know what is wrong with the cellphone in order to process the return. | |
| **Probability from Rationale-to-context Alignment Filter** | 0.48 |
| **Filtered Rationale** | |
| Subquestion 1: What is the reason for Person B's request to return the cellphone? (xIntent) | |
| Subanswer 1: Person B wants to return the cellphone because it is broken. | |
| Subquestion 2: What does Person A want to know about the broken cellphone? (oWant) | |
| Subanswer 2: Person A wants to know what is specifically wrong with the cellphone. | |
| Subquestion 3: What information might Person B provide to answer Person A's question? (xAttr) | |
| Subanswer 3: Person B might provide details about the specific issue with the cellphone, such as a cracked screen or malfunctioning buttons. | |

Table 14: Examples of rationales filtered out and passed the rationale-to-context alignment filter. The consistency scores are the outputted probability of the classifier.

| Context |  |
|---|---|
| A: How much for a bus pass? | |
| B: Well, for a monthly pass, it'll cost you $65. | |
| A: Is there anything else that doesn't cost as much? | |
| B: If you're a student, you can get a student bus pass. | |
| A: How much does a student pass cost? | |
| B: That actual bus pass is free. | |
| A: It doesn't cost anything? | |
| B: The only thing you'll have to pay for is the monthly sticker. | |
| **Response** | |
| A: Can you tell me how much that'll cost? | |
| **Probability from Rationale-to-Context Alignment Filter** | 0.99 |
| **Passed Rationale** | |
| Subquestion 1: What is the initial question of Person A to Person B? (xIntent) | |
| Subanswer 1: Person A wants to know the cost of a bus pass. | |
| Subquestion 2: What alternative option does Person B suggest to Person A? (oReact) | |
| Subanswer 2: Person B suggests that if Person A is a student, they can get a student bus pass. | |
| Subquestion 3: What additional cost is associated with the student bus pass? (xAttr) | |
| Subanswer 3: Person A wants to know the cost of the monthly sticker associated with the student bus pass. | |
| **Probability from Rationale-to-Context Alignment Filter** | 0.10 |
| **Filtered Rationale** | |
| Subquestion 1: What is Person A's question after learning about the student bus pass? (xIntent) | |
| Subanswer 1: Person A wants to know the cost of the monthly sticker for the student bus pass. | |
| Subquestion 2: What is the only cost associated with the student bus pass according to Person B? (xAttr) | |
| Subanswer 2: According to Person B, the only cost associated with the student bus pass is the monthly sticker. | |
| Subquestion 3: What might be the reason for Person A to ask about the cost of the monthly sticker? (oWant) | |
| Subanswer 3: Person A might want to compare the cost of the monthly sticker for the student bus pass with the cost of a regular monthly pass to determine the best option for them. | |

Table 15: Examples of rationales filtered out and passing the rationale-to-context alignment filter. The consistency scores are the outputted probability of the classifier.

| **Context** |
| --- |
| A: Hey, Mom. |
| B: Hi, sweetie. How was your day? |
| A: It was good. I went to the library and got some books. |
| B: That sounds like fun. What did you get? |
| A: I got a history book, a science fiction book, and a mystery book. |
| B: That sounds like a great mix! I'm sure you'll enjoy all of them. |
| A: I think so too. I can't wait to start reading them. |
| B: I'm glad you're enjoying your books. Reading is such a great way to learn new things and escape into new worlds. |
| A: Yeah, it is. I'm really looking forward to reading these books. Thanks for getting them for me, Mom. |
| **Response** |
| B: You're welcome, sweetie. I'm happy to help you find books that you'll enjoy. |
| **Passed Rationale** |
| Subquestion 1: What is Person A's reason for going to the library? (xIntent) |
| Subanswer 1: Person A went to the library to get some books. |
| Subquestion 2: How does Person B feel about Person A's book choices? (oReact) |
| Subanswer 2: Person B thinks that Person A's book choices are a great mix and that they will enjoy all of them. |
| Subquestion 3: What might Person B want to communicate to Person A about her role in helping Person A find enjoyable books? (oWant) |
| Subanswer 3: Person B might want to communicate to Person A that she is happy to help find books that Person A will enjoy. |
| **Filtered Rationale** |
| Subquestion 1: What is the intent of Person A in telling Person B about the books they got from the library? (xIntent) |
| Subanswer 1: Person A wants to share with Person B the books they got from the library. |
| Subquestion 2: How does Person B react to Person A's book choices? (oReact) |
| Subanswer 2: Person B is positive about Person A's book choices and encourages them to enjoy reading. |
| Subquestion 3: What might be Person B's motivation for helping Person A find books to enjoy? (xWant) |
| Subanswer 3: Person B might want to foster a love of reading in Person A and help them find enjoyment in learning and exploring new worlds through books. |

Table 16: Examples of rationales filtered out and passing the rationale-to-response alignment filter. The filtered rationale, although being related to the context, does not provide evidence specific to the response.

| Dialogue History | A: Hello. |
| | B: Hello. Is Anna there? |
| | A: No. I'm sorry Anna is out. |
| | B: Oh, well, could you please leave Anna a message for me? |
| | A: Sure. What is it? |
| | B: You know, Anna is coming for tea at my home. Would you ask her to see if she has my French dictionary? |
| | If she does, tell her to bring it with her. |
| **Ground-truth Response** | **A: All right. And where are you calling from?** |
| **Knowledge Source** | **Commonsense Knowledge & Generated Response** |
| None | Generated Response: |
| | A: Sure, I'll let Anna know to check for your French dictionary and bring it with her if she finds it. |
| Comfact | to help. |
| | to have a problem. |
| | to help someone. |
| | asks for help. |
| | |
| | Generated Response: |
| | A: I have a problem and I was hoping you could help me with it. |
| DIALeCT | The listener asked the speaker to leave anna a message on her behalf. |
| | The listener noted down the order of the speaker. |
| | Anna is a friend of the speaker. |
| | The speaker is curious to know whether anna is coming for dinner or not. |
| | The listener is happy to know that the speaker is sending her favourite book to her. |
| | |
| | Generated Response: |
| | A: Sure, I'll leave a message for her. By the way, do you know if Anna is planning to come for dinner tonight? |
| Reflect | Riley is responsible. |
| | Tracy might have gone out shopping for groceries. |
| | Alex might ask Anna to leave her house. |
| | Riley is a good friend. |
| | Sydney is a single woman. |
| | |
| | Generated Response: |
| | A: Sure, I'll let her know to check for your French dictionary when she gets back. |
| Self-CoT | Subquestion 1: What is the message that Person B wants to leave for Anna? (xIntent) |
| | Subanswer 1: Person B wants to ask Anna to check if she has their French dictionary and bring it with her if |
| | she does because Anna is coming for tea at Person B's home. |
| | Subquestion 2: What is the reason for Person B to ask Anna to bring the French dictionary? (oReason) |
| | Subanswer 2: Person B needs the French dictionary for some reason and wants Anna to bring it with her if she has it. |
| | Subquestion 3: What might be the potential impact if Anna forgets to bring the French dictionary? (oEffect) |
| | Subanswer 3: If Anna forgets to bring the French dictionary, Person B might not have access to it and might not be able |
| | to complete whatever task they needed it for. |
| | |
| | Generated Response: |
| | A: Okay, I'll let her know to check for the French dictionary and bring it with her if she has it. |
| DOCTOR | Subquestion 1: What is the request that Person B makes to Person A? (xIntent) |
| | Subanswer 1: Person B requests Person A to leave a message for Anna so that she can be informed about her visit to his home. |
| | Subquestion 2: What is the specific item that Person B asks Person A to ask Anna to bring with her? (xAttr) |
| | Subanswer 2: Person B asks Person A to ask Anna to bring his French dictionary with her. |
| | Subquestion 3: What might Person A need to clarify in order to fulfill Person B's request? (xNeed) |
| | Subanswer 3: Person A might need to clarify the address of Person B's home in order to deliver the message to Anna. |
| | |
| | Generated Response: |
| | **A: Of course, I'll leave a message for Anna to bring your French dictionary with her when she comes for tea at your** |
| | **home. Can you please remind me of your address so that I can make sure the message gets to her?** |

Table 17: An example of response generation using ChatGPT paired with different knowledge sources.

| | |
|---|---|
| **Dialogue History** | A: Do you want to go to the movies tonight?
B: I can't. I have to review my lessons.
A: Are you having a test tomorrow?
B: Yes. We're having our mid-term exam.
A: I wish you good luck.
B: Thanks. But I'm a little nervous.
A: Nervous? You used to study very well. |
| **Ground-truth Response** | B: But I haven't studied for a long time. |
| **Knowledge Source** | **Commonsense Knowledge & Generated Response** |
| None | Generated Response:
B: I know, but this exam is really important. I need to do well to keep my grades up. |
| Comfact | know what to do.
to find it.
to see what it is.
get's it.

Generated Response:
B: Great, thank you for the receipt. Let me take a look at the radio to see what the issue might be. |
| DIALeCT | The speaker's friend has to appear for a mid-term examination.
The listener booked a movie show tickets for the speaker and himself.
The speaker has already prepared for his mid term examination.
The speaker is encouraging the listener to study hard in the mid-term examination.
The listener is happy to hear that the speaker is going to watch him.

Generated Response:
B: I know, but this mid-term exam is really important. I don't want to take any chances. |
| Reflect | Riley is well-read.
Sydney might have been told by their teacher that they needed to review their lessons.
Sydney might go to a test.
Carson is careless.
Curious.

Generated Response:
B: I know, but I feel like I haven't studied enough this time. Plus, Carson keeps distracting me with his curiosity about everything. |
| Self-CoT | Subquestion 1: What is the reason for Person B's nervousness? (xReason)
Subanswer 1: Person B is nervous because they have a mid-term exam the next day.
Subquestion 2: What is Person A's opinion of Person B's studying abilities? (xAttr)
Subanswer 2: Person A believes that Person B used to study well.
Subquestion 3: What might be Person A's intention in reassuring Person B? (xIntent)
Subanswer 3: Person A might want to boost Person B's confidence and help them feel less nervous about the upcoming exam.

Generated Response:
B: Don't worry, you'll do great. Just remember to take deep breaths and stay focused. |
| DOCTOR | Subquestion 1: What is the reason for Person B's nervousness? (xAttr)
Subanswer 1: Person B is nervous because they have a mid-term exam to review.
Subquestion 2: What is Person A's observation about Person B's previous academic performance? (oReact)
Subanswer 2: Person A observes that Person B used to study very well.
Subquestion 3: What might be the reason for Person B's difficulty in understanding the material despite their previous studying efforts? (oCause)
Subanswer 3: The reason for Person B's difficulty in understanding the material might be due to the passage of time, as it may have affected their understanding of the material.

Generated Response:
B: I know, but it's been a while since I reviewed this material. I'm afraid I might have forgotten some important details. |

Table 18: An example of response generation using ChatGPT paired with different knowledge sources.

| Dialogue History | A: CFC Taxis.
B: Hello. I'd like to book a taxi, please.
A: Certainly. For what time?
B: Five o'clock tomorrow morning.
A: Where from, madam?
B: From Qilu Hotel.
A: Oh, I see. Where do you want to go?
B: To the railway station. How much will it cost?
A: About 20 dollars. |
|---|---|
| **Ground-truth Response** | B: **How long will it take to get there?** |
| **Knowledge Source** | **Commonsense Knowledge & Generated Response** |
| None | Generated Response:
B: Great, please book the taxi for me at 5 am tomorrow from Qilu Hotel to the railway station. Thank you. |
| Comfact | to be in touch.
to have a phone.
to talk to someone.
talk to someone.

Generated Response:
B: We usually talk on the phone once a week, but she's been really busy with work lately. |
| DIALeCT | The speaker asked the taxi driver about the cost of her train ticket.
The listener charged the speaker a fine for the extra time he'd like to travel.
The speaker has to reach the railway station on time.
The speaker is curious to know about the taxi fare.
The listener is happy to help.

Generated Response:
B: Great, thank you for letting me know. See you tomorrow morning at 5. |
| Reflect | Sydney is responsible.
Ash was on a business trip.
Sydney might hire a taxi.
Sydney is irresponsible.
Cameron is careless.

Generated Response:
B: Great, please book the taxi for me. Thank you. |
| Self-CoT | Subquestion 1: What is the intent of Person B when calling CFC Taxis? (xIntent)
Subanswer 1: Person B wants to book a taxi for a specific time and location.
Subquestion 2: What information does Person A need to know in order to book the taxi? (oNeed)
Subanswer 2: Person A needs to know the time and location of the pickup, as well as the destination.
Subquestion 3: What is the estimated cost of the taxi ride from Qilu Hotel to the railway station? (xAttr)
Subanswer 3: The estimated cost of the taxi ride is about 20 dollars.

Generated Response:
B: Would you like me to book the taxi for you now? |
| DOCTOR | Subquestion 1: What is the information that Person B wants to know after booking a taxi? (xWant)
Subanswer 1: Person B wants to know the arrival time of the taxi.
Subquestion 2: Why might Person B want to know the arrival time of the taxi? (oEffect)
Subanswer 2: Person B might want to plan the rest of the day accordingly based on the arrival time of the taxi.
Subquestion 3: What might be the reason for Person B to ask about the arrival time of the taxi? (xIntent)
Subanswer 3: Person B might want to make sure that the taxi arrives at the correct time and doesn't arrive too early or too late.

Generated Response:
B: **Great, thank you. Can you please let me know the estimated arrival time of the taxi?** |

Table 19: An example of response generation using ChatGPT paired with different knowledge sources.