# OpenReview forum: "Dialogue Chain-of-Thought Distillation for Commonsense-aware Conversational Agents"
_EMNLP/2023/Conference — EMNLP 2023 Main_

### Official Review · Reviewer_dhkb · 2023-08-07

**Soundness:** 3

**Excitement:**

4: Strong: This paper deepens the understanding of some phenomenon or lowers the barriers to an existing research direction.

**Paper Topic And Main Contributions:**

Identifying and aggregating evidence in chatbots is very challenging. In this paper they use counterfactual rationales to train the model.
They also use dialogue comprehension datasets such as DREAM to extract responses.


**Reasons To Accept:**

Figure 3 that it performs much better than common-sense annotation.

Table 5 also shows that the response of ChatGPT combined with the proposed DOCTOR is significantly better than baselines.





**Reasons To Reject:**

Abstract, last line, summarise the percentage improvement.

Pg 8, right col, line no 529, why are these approaches not applicable for commonsense reasoning?

Conclusion should provide the main contributions and percentage improvements.

How does this approach compare with ‘Augmenting end-to-end dialogue systems with commonsense knowledge’ AAAI 2018



**Reproducibility:**

4: Could mostly reproduce the results, but there may be some variation because of sample variance or minor variations in their interpretation of the protocol or method.

**Reviewer Confidence:**

4: Quite sure. I tried to check the important points carefully. It's unlikely, though conceivable, that I missed something that should affect my ratings.

---

> ### Author Rebuttal · Authors · 2023-08-29
>
> Thank you for the constructive feedback.
>
> **Q1 and Q3.**
> > Abstract, last line, summarise the percentage improvement.
>
> > Conclusion should provide the main contributions and percentage improvements.
>
>
> Thank you for your valuable comments on improving our paper. We will clarify our contribution in the camera-ready as you suggested:
> - We will add the following statement in the last line of the abstract: “We conduct extensive experiments to show that enhancing dialogue agents with high-quality rationales from DOCTOR significantly improves the quality of their responses. The results of the human evaluation show that 67% of responses using DOCTOR are preferred over the responses using knowledge from LLMs.”
>
> - We will revise the conclusion as follows: “Commonsense reasoning in conversations involves multi-hop reasoning, which poses challenges even for LLMs. To address this, we present a dialogue chain-of-thought distillation framework that selectively annotates high-quality rationales using LLMs. Our contributions are as follows: (1) With our framework, we collect DONUT, a large-scale dataset for dialogue CoT reasoning. (2) We present DOCTOR, a dialogue chain-of-thought reasoner trained on DONUT. (3)Through extensive experiments, we show the efficacy of DOCTOR, especially in the human evaluation, 67% of the responses using DOCTOR are preferred over the responses using knowledge from LLMs.”
> The revised version would be definitely clearer for readers to understand our contributions.
>
>
> **Q2. Pg 8, right col, line no 529, why are these approaches not applicable for commonsense reasoning?**
>
> We would like to clarify that while these approaches can be applied to commonsense reasoning, they focus on answering factoid questions and thus are suboptimal for commonsense reasoning in dialogues. We kindly note that reasoning in dialogues differs from reasoning in question answering in that the former involves a deeper understanding of the dialogue context (e.g., emotional state and situation of speakers) [1]. We apologize for any misunderstandings, and we will revise the sentence accordingly in our final draft to avoid any confusion.
>
>
>
>
> **Q4. How does this approach compare with ‘Augmenting end-to-end dialogue systems with commonsense knowledge’?**
>
> We first appreciate the reviewer for mentioning this paper, which is a foundational work on integrating commonsense knowledge into dialogue agents using commonsense knowledge graphs [2]. We will make sure that we include this reference in our final draft.
>
> We would also like to point out that while commonsense knowledge graphs provide valuable knowledge, integrating static knowledge graphs for conversations with vast world knowledge has always been a challenge [2, 3]. More recent studies thus rely on neural commonsense models [4], which offer more flexibility in representing and utilizing vast knowledge. Taking a step further, our work introduces a multi-hop commonsense reasoning model for dialogues that learns to represent more complex and diverse knowledge in dialogues using the reasoning ability of LLMs.
>
> [1] Sahand Sabour, Chujie Zheng, Minlie Huang, CEM: Commonsense-aware Empathetic Response Generation, AAAI 2022
>
> [2] Young, Tom, Erik Cambria, Iti Chaturvedi, Hao Zhou, Subham Biswas, and Minlie Huang, Augmenting end-to-end dialogue systems with commonsense knowledge, AAAI 2018
>
> [3] Christopher Richardson and Larry Heck. 2023, Commonsense reasoning for conversational ai: A survey of the state of the art, KnowledgeNLP-AAAI 2023
>
> [4] Bosselut, Antoine, Hannah Rashkin, Maarten Sap, Chaitanya Malaviya, Asli Celikyilmaz, and Yejin Choi, COMET: Commonsense Transformers for Automatic Knowledge Graph Construction, ACL 2019

---

### Official Review · Reviewer_kVfw · 2023-08-07

**Soundness:** 4

**Excitement:**

4: Strong: This paper deepens the understanding of some phenomenon or lowers the barriers to an existing research direction.

**Paper Topic And Main Contributions:**

This paper introduces a novel framework to model conversations, through multi-step reasoning using LLMs.  The challenge in modeling "deep" conversations is using commonsense reasoning at multiple turns of the conversations and collectively combine the reasoning steps to infer the main intent of the conversation.  This work demonstrates how "shallow" (single-step) approaches, such as integrating commonsense knowledge from per-trained KBs/models, and using single shot LLM prompting are not sufficient to tackle the task.
The authors contribute a 3 step process, first step is to extract implicit pairs of <question, answer> useful to model the conversation, using CoT approach.  The second step is to filter these conversations to ensure the sequence is "consistent".  The first 2 steps also allow to build a dataset to train the final conversational model in the third step, to generate the final dialogue, using implicit information from the previous steps.

**Questions For The Authors:**

Token based scores, Bleu or Rogue seems quite low, which makes sense for this challenging setting.  Wondering if you tried semantic based scoring approaches, such as BleuBERT?

**Reasons To Accept:**

The framework is novel and can be widely applied to nuanced conversations, or in general any text requiring deeper inference to understand the narrative.  Their method improves other competitive baseline approaches.

**Reasons To Reject:**

N/A

**Reproducibility:**

5: Could easily reproduce the results.

**Reviewer Confidence:**

4: Quite sure. I tried to check the important points carefully. It's unlikely, though conceivable, that I missed something that should affect my ratings.

---

> ### Author Rebuttal · Authors · 2023-08-29
>
> We truly appreciate your time and the fact that you recognize the value of our research.
>
> Following your suggestion, we conduct an evaluation using semantic-based metrics, BLEURT and BERTScore. We show the results in the following table. Thanks to your suggestion, we can provide more diverse criteria for evaluation not only based on lexical similarity but also semantic similarity. We will include these results in the camera-ready.
>
>
> | Method⠀⠀⠀⠀⠀⠀⠀⠀⠀⠀| \| |       |       | DailyDialog |          | \| |       |       | DREAM     |          | \| |       |       | MuTual    |          | \| |       |       | Reflect-9K |          |
> | ---------------- | --- | ----- | ----- | ----------- | -------- | --- | ----- | ----- | --------- | -------- | --- | ----- | ----- | --------- | -------- | --- | ----- | ----- | ---------- | -------- |
> |                  | \| | B-1   | R-L   | BERTScore   | BLEURT   | \| | B-1   | R-L   | BERTScore | BLEURT   | \| | B-1   | R-L   | BERTScore | BLEURT   | \| | B-1   | R-L   | BERTScore  | BLEURT   |
> | Cosmo            | \| | 20.04 | 13.69 | 0.8539      | \-1.0439 | \| | 20.98 | 14.03 | 0.8629    | \-1.0414 | \| | 17.63 | 12.03 | 0.8529    | \-0.9728 | \| | 14.97 | 11.80 | 0.8567     | \-1.0875 |
> | &plus;COMET w/ ComFact | \| | 19.54 | 14.05 | 0.8553      | \-0.9846 | \| | 20.90 | 14.09 | 0.8621    | \-1.0405 | \| | 17.56 | 12.41 | 0.8524    | \-0.9585 | \| | 15.80 | 12.09 | 0.8558     | \-1.084  |
> |&plus; DIALeCT          | \| | 19.63 | 14.31 | 0.8565      | \-0.9767 | \| | 20.69 | 13.82 | 0.8623    | \-1.0271 | \| | 18.01 | 12.23  | 0.854   | \-0.9738  | \| | 15.54 | 11.82 | 0.8559     | \-1.1007 |
> | &plus;Reflect          | \| | 19.44 | 13.58 | 0.8529      | \-1.0515 | \| | 18.23 | 13.30 | 0.8569    | \-1.0969 | \| | 18.57 | 12.01 | 0.8438    | \-1.0441 | \| | 15.33 | 12.05 | 0.8565     | \-1.099  |
> | &plus;DOCTOR           | \| | 20.43 | 14.68 | 0.8572      | \-0.9593 | \| | 21.26 | 14.26 | 0.8632    | \-1.0215 | \| | 17.90 | 12.35 | 0.8539    | \-0.9737 | \| | 16.66 | 12.11 | 0.8569     | \-1.0749 |
> |                  | \| |       |       |             |          | \| |       |       |           |          | \| |       |       |           |          | \| |       |       |            |          |
> | ChatGPT          | \| | 17.25 | 14.72 | 0.8493      | \-0.9676 | \| | 18.90 | 14.86 | 0.8642    | \-0.9983 | \| | 17.92 | 14.83 | 0.8556    | \-0.8589 | \| | 17.28 | 12.77 |      0.8619      |   -0.9737       |
> | &plus;COMET w/ ComFact | \| | 18.24 | 14.56 | 0.8501      | \-0.9341 | \| | 20.09 | 14.44 | 0.8642    | \-0.9805 | \| | 19.32 | 15.46 | 0.8567    | \-0.8325 | \| | 17.38 | 13.28 |   0.8611         |    -0.9694      |
> | &plus;DIALeCT          | \| | 16.61 | 13.55 | 0.8471      | \-1.0129 | \| | 18.00 | 13.16 | 0.8615    | \-1.0492 | \| | 19.15 | 15.55 | 0.8574    | \-0.8398 | \| | 17.48 | 12.96 |       0.8602     |  -1.0101       |
> | &plus;Reflect          | \| | 17.47 | 13.80 | 0.8474      | \-1.0365 | \| | 19.02 | 13.37 | 0.8612    | \-1.0728 | \| | 18.14 | 14.27 | 0.8579    | \-0.8325 | \| | 18.24 | 12.54 |    0.8600        |   -0.9702       |
> | &plus;Self-CoT         | \| | 18.16 | 12.62 | 0.8461      | \-0.9953 | \| | 18.88 | 12.17 | 0.859     | \-1.0544 | \| | 19.97 | 13.71 | 0.8549    | \-0.8502 | \| | 14.53 | 11.56 |      0.8579      |  -1.0274        |
> | &plus;DOCTOR           | \| | 19.61 | 15.63 | 0.8512      | \-0.9224 | \| | 21.2  | 14.93 | 0.8654    | \-0.965  | \| | 20.19 | 15.55 | 0.8583    | \-0.7906 | \| | 18.54 | 12.85 |     0.8621       |-0.9649|

---

### Official Review · Reviewer_cf7H · 2023-08-11

**Soundness:** 4

**Excitement:**

4: Strong: This paper deepens the understanding of some phenomenon or lowers the barriers to an existing research direction.

**Paper Topic And Main Contributions:**

The paper proposes the use of CoT methodology for generating appropriate responses to a dialogue. First, the authors propose the Donut dataset that contains multiple CoT rationales for generating appropriate responses. The CoT rationales come in the form of QA pairs which an LLM can utilise to generate appropriate and relevant response. Then, the authors propose the Doctor framework, that utilises this large-scale Doctor dataset for supplementing the LLM with CoT rationales and thus generation improved responses.

Researchers working on dialogue system can benefit from the utilisation of Donut dataset for different dialogue tasks. Additionally, the authors have showcased the strong performance of Doctor on multiple dialogue dataset.

**Questions For The Authors:**

See the weakness.

**Reasons To Accept:**

1. The paper is well written, easy to read and understand.
2. I appreciate the authors for doing a robust evaluation of the Donut dataset as this would be helpful for other researchers working on dialogue system.
3. The simple yet novel use of rationales for generating relevant responses using the Doctor framework is appreciable and is evidence through strange empirical performance on multiple dialogue dataset.

**Reasons To Reject:**

1. The critic model is used to determine if a rationale is valid or not. It would be good to have an understanding of the number of training instances required for training the critic model.
2. An analysis on the number of rationales required for generating a relevant response would be helpful.

**Reproducibility:**

4: Could mostly reproduce the results, but there may be some variation because of sample variance or minor variations in their interpretation of the protocol or method.

**Reviewer Confidence:**

4: Quite sure. I tried to check the important points carefully. It's unlikely, though conceivable, that I missed something that should affect my ratings.

---

> ### Author Rebuttal · Authors · 2023-08-29
>
> We thank you for acknowledging the importance of our work and providing helpful suggestions.
>
> **Q1.** For our dataset, we sampled 6K dialogues from SODA. From these, we constructed 6K $(U_{<t}, U_t, \tilde{Z})$ pairs and another 6K $(\tilde{U}_{<t}, U_t, \tilde{Z})$ pairs, resulting in a total of 12K samples. We then randomly divided these samples into training, validation, and test sets, with 10K, 1K, and 1K samples respectively. We detailed the training of our critic model in Appendix A.2. Thanks to your questions, we recognized the importance of this information and believe it would be beneficial to include it within the main 8 pages of the paper. Again, we appreciate your keen observation.
>
> **Q2.** We agree with the reviewer that the number of reasoning steps $k$ in the rationales could be a factor that affects the quality of the generated rationales and the responses. From a qualitative analysis on the generated rationales, we discover that rationales shorter than 3 steps often lack sufficient evidence to support the reasoning while rationales longer than 3 steps tend to aggregate excessive evidence and contain spurious reasoning. Thus, we initially select the $k$ as 3.
>
> To better understand the effect of $k$ on the quality of rationale, we conduct human evaluation using 100 random dialogue samples from DailyDialog, DREAM, and MuTual. For each dialogue, we prompt ChatGPT to generate five CoT rationales with $k=\\{1,2,3,4,5\\}$, respectively, as we do in Section 3.1. Using the same criteria from Table 3, we asked 3 different workers from Amazon Mechanical Turk to evaluate the quality of the rationale from each dialogue.
>
> | $k$ | Consistency | Helpfulness | Specificity | Overall |
> | --- | :-----------: | :-----------: | :-----------: | :-------: |
> | 1   | 78.1        | 77.9        | 83.4        | 79.9    |
> | 2   | 87.5        | 78.4        | 81.1        | 80.7    |
> | 3   | 91.2        | 81.4        | 86.9        | 87.1    |
> | 4   | 88.5        | 78.1        | 88          | 83.6    |
> | 5   | 86.9        | 76.5        | 83.2        | 82.6    |
>
> In line with our initial motivation on choosing $k$, the workers preferred the rationales with $k=3$ most in terms of consistency, helpfulness, and overall. The Krippendorff alpha (0.82, 0.58, 0.74, 0.71) scores show a moderate agreement among the raters. We will include these results in the camera-ready version to provide empirical evidence for our choice of $k$. We once again thank you for your insight.

---

### Official Review · Reviewer_FRrp · 2023-08-12

**Typos Grammar Style And Presentation Improvements:** NA
**Soundness:** 4

**Excitement:**

4: Strong: This paper deepens the understanding of some phenomenon or lowers the barriers to an existing research direction.

**Missing References:**

NA

**Paper Topic And Main Contributions:**

The paper tries to improve common sense reasoning over multiple turns in conversational agents by augmenting with a conversation rationale produced by a large language model. They propose to overcome the limitations of LLMs to generate reliable rationales by training a reliable COT reasoner on selective rationales.

**Questions For The Authors:**

A. In eq. 4, line 207, if Z is a helpful rationale, P(u_t|Z, U_{<t}) should be greater than P(u_t| U_{<t}) hence \tau = 1 should be the case. Why is \tau taken to be 0.95?

B. In lines 188 to 192, (a critic model to detect counterfactual rationales generated without conditioning on the dialogue context. We ask the LLM to generate a counterfactual rationale \tilde{Z} from a counterfactual context \tilde{U}_{<t} containing only the last utterance),  "without conditioning on the dialogue context" and "from a counterfactual context tilde{U}_{<t} containing only the last utterance" is not clear and seems contradictory. Can you please explain?

C.  What are the cases where the LLM fails to generate an aligned rationale and the cases where despite generating an aligned rationale, the chat LLM fails to produce a coherent response? Can you find some broad classes of conversations where your framework fails?

**Reasons To Accept:**

1. The paper addresses a very important problem of common sense reasoning over multiple turns in a conversation and provides a reliable dialogue rationale dataset that might be useful to the community.

2. The paper is very well written, precise, and easy to follow.

**Reasons To Reject:**

Although the paper proposes an overall novel and robust framework, the individual components have been already explored by the community.

**Reproducibility:**

5: Could easily reproduce the results.

**Reviewer Confidence:**

4: Quite sure. I tried to check the important points carefully. It's unlikely, though conceivable, that I missed something that should affect my ratings.

---

> ### Author Rebuttal · Authors · 2023-08-29
>
> We appreciate your comments and feedback. We address the following important questions.
>
>  **Question A.** Why is $\tau$ taken to be 0.95 rather than 1?
>
> We strongly agree that $\tau = 1$ might be the reasonable choice for our hyperparameter in the rationale-to-response alignment filter. However, in the filtering process, we observed a gray area around $\tau=1$ where a considerable amount of samples show only subtle differences in the helpfulness ratio, implying that their detrimental effects might be trivial. Our motivation was that adding such samples would result in a larger dataset with more variation in annotated rationales, thus enhancing the performance and robustness of the commonsense reasoner.
>
> We tested various values around 1 for $\tau$ (0.9, 0.95, 1, and 1.05) to study this effect and find a balance between the quality and quantity of the annotated rationales. We ran experiments using ChatGPT as a dialogue agent tested on DailyDialog. The results are shown in the table below:
>
> | $\\tau$     | B-1   | B-2  | B-4  | R-L   | \|\| | Relative size compared to DONUT |
> | ----------- | ----- | ---- | ---- | ----- | ------ | ------------------------------- |
> | 0.9         | 18.75 | 7.73 | 2.26 | 14.69 | \|\| | 1.1                             |
> | 0.95 (Ours) | 19.61 | 8.44 | 2.69 | 15.63 | \|\| | 1                               |
> | 1           | 19.58 | 8.34 | 2.6  | 15.52 | \|\| | 0.47                            |
> | 1.05        | 19.15 | 8.13 | 2.51 | 15.19 | \|\| | 0.19                            |
>
> We find that $\tau = 0.95$ shows the best results, thus we chose this value as a hyperparameter for $\tau$. We should’ve included these results in our paper to show the readers the reason behind the choice of $\tau$. Thank you again for pointing it out, we will include them in the camera-ready version.
>
> **Question B.**
>
> Counterfactual rationales are generated while being conditioned only on the last utterance ($u_{t-1}$) preceding the target response rather than the entire dialogue context $U_{<t}$. The sentence in line 189-190 should be revised as “a critic model to detect counterfactual rationales generated without correctly grounding on the dialogue context.”. We will clarify this in the camera-ready.
>
> **Question C.**
>
> Thank you again for your valuable question. We believe that analyzing these cases will definitely help us to gain insights regarding improving our work.
> To find failure cases, we randomly select 600 samples from DailyDialog and ask workers from AMT to evaluate rationales and generated responses by answering the following yes-no questions:
>
> - Do you agree that knowledge is well-aligned with the dialogue context?
> - Do you agree that knowledge is well-aligned with the reference response?
> - Do you agree that the predicted response is coherent with the dialogue context?
>
> Each sample is evaluated by 3 different workers to reduce variance and improve the reliability of the evaluation. Please refer to the table in C-3. Statistics for the statistics from the workers’ evaluation.
>
> **C-1. When does DOCTOR fail at generating aligned rationales?**
>
> We manually inspect samples that at least 2 workers consider not aligned with the dialogue context or the response. We categorize the type of failure and describe the cases where DOCTOR fails at generating aligned rationales.
>
> **C-1-1. Failed cases of rationale-to-context alignment:**
>
> We observe two major types of cases where DOCTOR hallucinates and generates rationales that are not aligned with the dialogue contexts.
>
> - *Complex dialogue flow (49%):* We find that the questions in CoT rationale are often well generated. However, DOCTOR struggles to follow the dialogue flow and thus fails to answer the questions correctly. This results in failure to aggregate enough evidence.
>
> - *Lack of evidence in dialogue context (38%):* DOCTOR concludes with a statement that is unrelated to what is mentioned in the dialogue context or cannot be induced based on the dialogue context when generating the rationales. This is mainly due to the absence of necessary evidence for reasoning, e.g. when a dialogue context is too short to contain enough evidence that supports coherent reasoning.
>
> **C-1-2. Failed cases of rationale-to-response alignment:**
>
> We find two major reasons that lead to rationales misaligned with the reference response.
>
> - *Openness of dialogue (33%):* Due to the open-ended nature of dialogues, many possible responses exist for a single dialogue context. Therefore, while the rationales are plausible enough, they lead to responses different from the reference responses.
>
> - *Small talks (31%):* DOCTOR often generates sophisticated rationales to describe its reasoning even in scenarios where simple conversations are enough. e.g., daily greetings.
>
> **C-2. When does the chat LLM fail to produce a coherent response even if it is provided with an aligned rationale by DOCTOR (given the dialogue context)?**
>
> We observe that it is highly unlikely that the response will be incoherent if the provided rationale is well aligned, considering that out of 81.1% of samples with aligned rationales, only 2.1% are deemed incoherent.
>
> One possible reason behind these few failure cases is that rationales generated by DOCTOR might be too complex and lengthy due to the complex nature of dialogue. In such cases, chat LLMs sometimes fail to fully reflect the rationales in their responses, leading to incoherent responses.
>
>
>
>
>
>
> **C-3. Statistics:**
> |                             | Successful r-to-c alignment | Failed r-to-c alignment |
> | --------------------------- | :---------------------------: | :-----------------------: |
> | Successful r-to-r alignment | 81.2%                       | 1.5%                    |
> | Failed r-to-r alignment     | 12.3%                       | 4.0%                    |
>
>
> |                     | Aligned rationale | Not aligned rationale |
> | ------------------- | :-----------------: | :---------------------: |
> | Coherent response   | 79.0%             | 11.8%                 |
> | Incoherent response | 2.1%              | 7.0%                  |

---

### Meta-Review · Area_Chair_QRsr · 2023-09-20

**Recommendation:** 5

**Metareview:**

The authors tackle the problem of multi-hop common sense reasoning in dialogue and propose a dataset and a model. The reviewers appreciated the dataset, the clarity of the work, and the results achieved. They did raise a few concerns such as the fact that the novelty lies in the combination of known methods or some missing details but the authors adequately address them. Therefore I recommend accepting this paper.

---

### Decision · Program_Chairs · 2023-10-07

**Decision:**

Accept-Main

**Comment:**

The authors tackle the problem of multi-hop common sense reasoning in dialogue and propose a dataset and a model. The reviewers appreciated the dataset, the clarity of the work, and the results achieved. They did raise a few concerns such as the fact that the novelty lies in the combination of known methods or some missing details but the authors adequately address them. Therefore I recommend accepting this paper.